



# Exploring the provenance of information across Canadian hydrometric stations: Implications for discharge estimation and uncertainty quantification

Shervan Gharari[1], Paul H. Whitfield[2], Alain Pietroniro[3], Jim Freer[2], Hongli Liu[4], Martyn P. Clark[5]

[1]Centre for Hydrology, University of Saskatchewan, Saskatoon, Saskatchewan, Canada.
[2]Centre for Hydrology, University of Saskatchewan, Canmore, Alberta, Canada.
[3]Schulich School of Engineering, University of Calgary, Calgary, Alberta, Canada.
[4]Department of Civil and Environmental Engineering, University of Alberta, Edmonton, Alberta, Canada.
[5]Department of Geography and Planning, University of Saskatchewan, Saskatchewan, Canada.

**Key Points:**

- The Water Survey of Canada's standard operating procedures in estimating discharge from stage values are explored and explained.
- Given standard operating procedures, four major discharge and uncertainty estimation categories were identified using a standalone Python workflow.
- 67% of the reported discharge values in the operational database could be explained following the concept of rating curves and temporary shifts.
- Users of hydrometric datasets are encouraged to understand the provenance of that data, and its fitness for purpose, alongside spatial and temporal differences in uncertainty.

Corresponding author: Shervan Gharari, shervan.gharari@usask.ca



## Abstract

Accurate discharge values play a critical role in water resource planning and management. However, it is common for users, modelers, and decision-makers to consider these values as true and deterministic, despite the subjective and uncertain nature of the estimation process. To address the issue, this study was conducted to identify the discharge estimation methods and associated uncertainties of hydrometric measurements in Canada. The study involved an exploration of multiple operating procedures for rating curve construction and discharge estimation across 1800 active Water Survey of Canada (WSC) hydrometric stations using an independent workflow. The first step involved understanding the discharge estimation process used by the WSC and the standard operating procedures (SOP) for inferring discharge from stage measurements. During the implementation of the workflow, it was observed that manual intervention and interpretation by hydrographers were required for time-series sequences labeled as "override" and/or "temporary shift". The workflow demonstrated that 67 % of existing records could be adequately recreated following the rating curve and temporary shift concept, while 33 % followed the other discharge estimation methods (override). Novel methods for discharge uncertainty estimation should be sought given the practices of override and temporary shift by the WSC. This study attempts to reconcile the significant issue of estimating uncertainty in published discharge values, particularly in the context of open science and Earth System modeling. By collaborating with the WSC, this research aims to improve the understanding of the processes used for discharge estimation and promote wider access to metadata and measurements for more accurate uncertainty quantification.





## Plain Language Summary

This study provides insight into the practices that are incorporated into discharge estimation across the national Canadian hydrometric network operated by the Water Survey of Canada, WSC. The procedures used to estimate and correct discharge values are not always understood by end-users. Factors such as ice cover, and sedimentation limit the ability of accurate discharge estimation. Highlighting these challenges sheds light on difficulties in discharge estimation and associated uncertainty.



# 1 Introduction

River discharge or streamflow has significant importance for planning, impact and sustainability assessment, and Earth System modeling (McMillan et al., 2017; Shafiei et al., 2022). River discharge is the integration of other fluxes such as precipitation, evaporation, and soil moisture level at catchment- and basin-scale and hence carries important information about the natural and anthropogenic processes. Given this importance, the national gathering of river discharge data is typically a data product that governments provide as basic national infrastructure to support decision-making, planning, and water management objectives of governments, industry, and private sectors.

River discharge values are typically obtained by using a relationship called rating curve (Rantz, 1982) to convert measurements of stage values (water level) to estimates of discharge (water volume over time). The direct discharge measurements are made using velocity measurement techniques such as velocity/flow meters, Acoustic Doppler systems, or other techniques. Each measurement technique, device, frequency, and rule result in various error magnitudes (Pelletier, 1989). Rating curves are developed through occasional discharge measurement activities in the field, where hydrographers relate those direct measurements to river stages. The structure of the residuals model for rating curves can then be characterized by comparing measurements to rating curves. The residuals model can then be used, often in a straightforward way, to estimate discharge uncertainty from continuous stage measurement (Whalley et al., 2001; Cohn et al., 2013; Coxon et al., 2015; Huang, 2018; Kiang et al., 2018).

In addition, errors in discharge values also stem from the (limited) capability of rating curves to represent time-dependent changes in stage-discharge relationships. Such time-dependent changes in river conditions come from local hydrodynamics and environmental conditions. This includes time-dependent changes in river conditions that introduce backwater effects due to sedimentation, and vegetation growth or ice formation, amongst others. The stage-discharge relationships defined by rating curves are generally functional forms (single curve) while in reality, they may be hysteretic due to the dynamic nature of water movement in the channel (Tawfik et al., 1997; Wolfs & Willems, 2014; Lloyd et al., 2016; Gharari & Razavi, 2018). For example, the rising limb and falling limb of a flood hydrograph may exhibit different discharge values for the same stage. This difference between the assumed stage-discharge relationship and the dynamic nature of the stage-discharge relationship is a source of uncertainty (among many other sources of discharge uncertainty).

Lastly, *standard operating procedures* or SOPs that are developed and used by hydrometric agencies for translating water level to discharge are often established for constant re-assessment. In many instances, the stage-discharge relationship can be subject to the hydrographers' intervention. As an example, the process of creating a rating curve from observational discharge measurement may need to follow agreed-upon institutional or organizational procedures. In addition, updating rating curves over time, to try to maintain the accuracy of relationships, may result in more challenges in uncertainty quantification associated with the rating curve.

Given the differences in operating procedures, separating the above sources of uncertainty quantitatively is challenging and needs an extensive understanding of the operating procedures to determine the magnitude of each of the sources of uncertainty. Despite this difficulty, the communication of the discharge uncertainty is becoming increasingly important as hydrological, water quality, and water management models, which are often used for decision-making, are based on these published and approved estimates of river discharge.

The study's ultimate goal is to assist with the quantification of uncertainty in the discharge measurements taken at Canadian hydrometric stations. The study seeks to identify critical decisions at the WSC's quality assurance and management system (QMS) to aid in this process. The study is a necessary step in diagnosing the issue of discharge uncertainty estimation in Canadian hydrometric stations. The study seeks to answer the following questions:



- What are the standard operating procedures followed by hydrographers at the WSC for discharge estimation?
- What are the critical decisions at the WSC that affect discharge estimation and associated uncertainties and how they can be categorized?
- How can access to metadata and measurements be improved to aid in the estimation of discharge uncertainty for Canadian hydrometric stations?

This paper is organized as follows. First, the terminologies are introduced to familiarize readers with the institutions, SOPs, concepts used in this study, and the workflow from data acquisition to river discharge estimation. This is followed by the results section where examples of rating curves and their relationship to observations of stage-discharge values are discussed. The estimated discharge values by WSC are reproduced using the available stage values and information in the production system. The paper concludes by discussing the findings and suggestions for essential data acquisition and archiving that will allow for better uncertainty estimation for Canadian hydrometric stations.

## 2  Data, Terminologies, and Methodologies

### 2.1  Canada's hydrometric monitoring program

Canada like many other nations has invested heavily in its national hydrometric monitoring program through the Water Survey of Canada, WSC, and in the publicly available national service and historic discharge records (refer to Table-1 for terminologies that are used in this work). WSC is a unit of the National Hydrological Service for Canada which is housed within the Canadian Government and is part of the Federal Department of Environment, known as Environment and Climate Change Canada (ECCC). WSC, an ISO 9001-certified organization, oversees the collection, harmonization, and standardization of discharge information in a cost-shared partnership with provincial and territorial governments across Canada. WSC divides its data into 5 regional entities: (1) Pacific and Yukon Region (British Columbia and Yukon), (2) Prairie and Northern Region (Alberta, Manitoba, Saskatchewan, Northwest Territories, and Nunavut) (3) Ontario Region, (4) Québec Region, (5) Atlantic Region (New Brunswick, Newfoundland, and Labrador Nova Scotia, and Prince Edward Island). The Ministère de l'Environnement et de la Lutte contre les changements climatiques operates the majority of the Quebec hydrometric stations and contributes these data to the national database under the cost-share agreements and partnerships. Other provinces, also operate their stations and contribute to the network. WSC monitoring stations include measurements in real-time of water levels in lakes and rivers and real-time river discharge estimation for the majority of its active stations. WSC, currently, operates approximately 1800 active stations across Canada with its partner for discharge estimation. The number of active stations has changed over time while some historical stations are discontinued (not active currently). Detailed descriptions of the history of the WSC, its partnership, and technical evolution are documented (Halliday, 2008; Kimmett, 2022).

### 2.2  Overview of Current Production System

WSC uses the Aquarius™ operation system maintained and operated by Aquatics Informatics. Aquarius™ is used for interaction with the operational database and manipulation of values for discharge estimation. This system was tailored to the WSC SOPs and QMS, and has been in use since 2010. The Aquarius™ system allows for real-time water level reporting and flow data estimations for most WSC stations equipped with telemetry systems. These stage values go through automated checks to account for faulty readings. Meanwhile, WSC hydrographers may perform discharge activity and enter the measured discharge values into the system. The estimated discharge may then be used





to correct based on discharge measurements, depending on conditions. The hydrogra-
pher might decide to apply or change previously estimated discharge values based on dis-
charge measurements and other environmental factors or move on with testing a new rat-
ing curve. Aquarius™ including its graphical user interface or GUI, provides many op-
tions to hydrographers to revise the discharge values, smooth discontinuities, and fill gaps
among others. These provisional data are later quality assured and approved using a rig-
orous approval process. The aggregated discharge values at daily temporal resolution are
disseminated publicly through the National Water Data Archive of Canada called HY-
DAT.

The most important and easily measured variable in hydrometry is *stage* or *wa-*
*ter level*. The accurate measurement of stage values is crucial as it is the main variable
used in combination with the rating curve to estimate discharge. The recorded stage val-
ues are at temporal resolutions programmed into the field-based logger system and are
typically in the order of minutes. It is noteworthy to mention that the stage logger time
steps are currently set at 5 minutes, in the past, the observation of the stage values would
vary between sites and be recorded as daily, half-daily, hourly, or quarter-hourly depend-
ing on the station. Therefore the stage time series might have various temporal resolu-
tions over the long-term historical record.

Discharge values are also reported at temporal logger resolution in the production
database. The reported discharge values are accompanied by quality assurance flags that
identify the condition under which the river discharge is estimated (explained in Table-
1). There is information in the production database regarding *field visits* which include
checking of the instruments or *stage-discharge measurements* that includes the direct mea-
surement of river discharge using techniques such as *mid-section*, using standard flow-
meters, or *Acoustic Doppler* equipment. In practice, multiple discharge measurements
are made to determine a consistent flow estimate, particularly when the measured dis-
charge deviates substantially from the expected discharge estimate derived from the rat-
ing curve (stage-discharge relationship). The discharge measurement activities are es-
sential to confirm or adjust rating curves.

The earliest records of stage values, in the current WSC operational database, are
from the mid-1990s. These data were transferred from the previous newleaf production
system when Aquarius™ was first introduced. The reader should note what is contained
in the operational database is only a fraction of the existing historical time series that
exists in various forms at WSC regional offices or earlier database systems. For exam-
ple, for the Bow River at Banff station located in the province of Alberta, the stage and
associated estimated discharge records start from 1995 in the operational database while
the reported discharge in the HYDAT dataset goes back to 1909. Similarly, the earli-
est records of observational field discharge measurements and the earliest rating curve
recorded for each station in the operational database extend mostly to the 1970s and 1980s.
For the same station, the existing rating curves in the operational database system be-
gin in 1990, despite over 100 years of record. Earlier rating curves cannot be accessed
from the operational database as they have not been transferred into this system, how-
ever, all records are available, many in hard copies in the WSC regional offices. This is
a similar story for historical field discharge measurements; not all the earlier historical
observations have been carried over to the current operational database. Again, for the
Bow River at Banff station, the earliest observational discharge in the operational database
is from 1986. The difference between the period of the digital operational database ac-
cessible by Aquarius™ and records that exist at WSC regional offices needs to be empha-
sized since the present analysis is limited to data that is contained in the current oper-
ational database.

The focus of this study is only on active stations. Each station is defined by a *sta-*
*tion ID*. The station ID is a unique identifier for each hydrometric station and its ap-
proximate location using a standard WSC naming convention. In this convention, the
first two digits define the major drainage basin in which the station is located (01-11,
see Figure-1). The two digits are followed by two letters that define the location of sub-



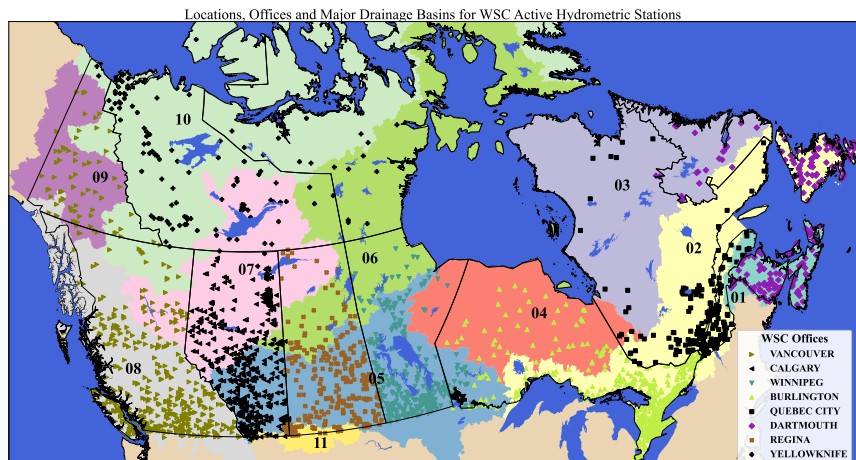

Figure 1: Location of active hydrometric stations across Canada. The eleven major drainage basins are (01) Maritime Provinces, (02) St. Lawrence, (03) Northern Quebec and Labrador, (04) Southwestern Hudson Bay, (05) Nelson River, (06) Western and Northern Hudson Bay, (07) Great Slave Lake, (08) Pacific, (09) Yukon River, (10) Arctic, and (11) Mississippi River. These digits are the first two characters in station IDs. The province of Quebec stations that are operated by Ministère de l'Environnement et de la Lutte contre les changements climatiques of the Province of Québec are not included in the WSC production database, nor are stations operated by other government agencies, crown or private corporations.

basins ordered from headwaters to the mouth in each major drainage basin (AA, BA, BB, BC, etc). The ID ends with a three-digit sequential number of the station in sub-basins. As an example, the station ID of Bow River at Banff, 05BB001, indicates it was the first station in sub-basin BB that is located in Saskatchewan/Nelson River basin identified by the leading code of 05.

### 2.3 Rating Curves

Rating Curves are perhaps the most commonly used method for river discharge estimation derived from stage observations. Rating curves are functional hydraulic relationships that relate river stage values to discharge values. In the WSC operational database, each rating curve is tied to an effective period, from a start to an end date, where the rating curve is considered the valid expression to estimate discharge values from stage records. Rating points are pairs of stage and discharge values that define the form of the rating curve functions (red points on Figure-2a,b). For the interpolation between the two consecutive rating curve points, the Water Survey of Canada uses two major approaches: (1) *linear table* (2) *logarithmic table*. In a linear table, a linear relationship is assumed between the rating points (Figure-2a), while in a logarithmic table, a logarithmic relationship is used instead (Figure-2b). The logarithmic relationship is defined by the form of $Q_t = a(H_t - O)^b$ with parameters $a$ and $b$ and an offset value of $O$. The offset values are archived alongside the rating points in the production system database while $a$ and $b$ can be inferred using the position, read stage, and discharge, of the consecutive rating curve points. $H_t$ is the measured stage and $Q_t$ is estimated discharge at time $t$. The logarithmic expression of rating curved resembles the hydraulic equations relating



Table 1: General definitions

| Item | Description | Unit |
|---|---|---|
| ECCC | Environment and Climate Change Canada is the department of the Government of Canada responsible for coordinating environmental policies and programs. | [-] |
| WSC | The Water Survey of Canada, part of ECCC, is responsible for maintaining hydrometric stations across Canada and reporting the discharge values for each hydrometric station. | [-] |
| Regions | The Water Survey of Canada is divided into five regions (1) Pacific and Yukon Region (British Columbia and Yukon), (2) Prairie and Northern Region (Alberta, Manitoba, Saskatchewan, Northwest Territories, and Nunavut) (3) Ontario Region, (4) Québec Region, (5) Atlantic Region (New Brunswick, Newfoundland, and Labrador Nova Scotia, and Prince Edward Island). | [-] |
| WSC [regional] offices | Offices of the Water Survey of Canada, also known as regional offices, are responsible for nearby stations and house hydrographers and equipment | [-] |
| Major drainage basins | Major drainage basins are described by a code from 01 to 11; these basins are (01) Maritime Provinces, (02) St. Lawrence, (03) Northern Quebec and Labrador, (04) Southwestern Hudson Bay, (05) Nelson River, (06) Western and Northern Hudson Bay, (07) Great Slave Lake, (08) Pacific, (09) Yukon River, (10) Arctic, and (11) Mississippi River. | [-] |
| Standard operation procedures or SOPs | The agreed-upon procedures followed at WSC for discharge estimation. | [-] |
| Operational or production database | The database that includes the time series of various variables and their metadata. | [-] |
| Aquarius™ | The system that facilitates the interactions with operational databases such as collection and archiving of data for hydrometric stations and associated workflows and standard operating procedures, SOPs, for discharge estimation. | [-] |
| API or application programming interface | The system which allows reading and interrogation of the operational database, outside of Aquarius™, using requests and responses from the server where the operational database is located. | [-] |
| HYDAT | Publicly available dataset that includes historical daily discharge values for Canadian hydrometric stations. | [-] |
| Station ID | The Station ID is encoded based on the major drainage basins in which it is located (01 to 11) and the basins and sub-basins (e.g. AA - AZ approximately from head to mouth) and a sequential number (001 - 999) resulting in a Station ID such as 01AA001. | [-] |
| Stage | Stage is the measured water level height of the free surface of a river. Stage values are reported at the given time based on the frequency such as daily, hourly, or quarter-hourly, etc. | [m] |
| River discharge or streamflow | The flow of water at a cross-section of a river. Normally reported in cubic meters per second which is the product of a velocity ($[\text{m s}^{-1}]$) and a cross-sectional area ($\text{m}^2$). | $[\text{m}^3\ \text{s}^{-1}]$ |
| Flags | Flags (SYM or symbol in HYDAT dataset, grade code in operational database) that define the condition of inferred reported discharge. The flags are E - Estimate, A - Partial Day, B – Backwater conditions including ice condition, D - Dry, and R – Revised | [-] |
| Field visits | Any type of field activity that involves a visit to the station by operators or hydrographers. This may include reporting the current technical parameters such as equipment, batteries, and power, or observation of the condition of the river section such as the presence of ice, backwater, etc (while excluding stage-discharge measurements). | [-] |
| Discharge activities or field discharge measurement | Refer to an activity in which hydrographers measure discharge and its associated stage. | [-] |
| Active stations | The stations that are currently in operation and collect data (in contrast to discontinued stations). | [-] |

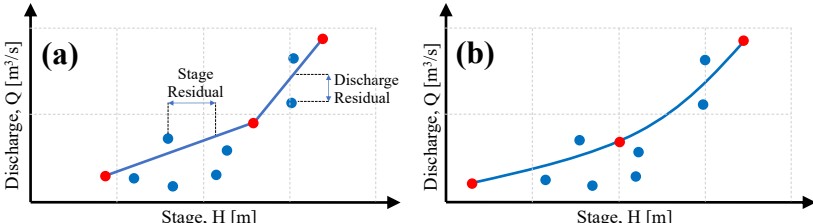

Figure 2: Examples for (a) linear table, and (b) logarithmic table rating curves. The blue points are the observation points of the measured stage and discharge during discharge activities; the rating points that define the rating curve are shown in red. In practice, these are not equations describing curves but lookup tables that record stage and discharge values.

water elevation to discharge. The offset, $O$, can also be referred to as reference elevation
or $H_0$ and alongside parameter $a$ and $b$ can reflect "hydraulic" characteristics (Reitan
& Petersen-Øverleir, 2011).

### 2.4 Managing Rating Curves Changes

The process of managing changes that affect a rating curve can be broken down
into three major practices, which are defined in the Water Survey of Canada (WSC) Stan-
dard Operating Procedures (SOPs). These changes can include non-functional relation-
ships such as hysteresis, or non-stationary relationships over time due to physical and
environmental factors. The processes are itemized below.

- **[Re]construction of rating curves:** New observations that indicate a change
  to the local hydraulic realities may require an establishment of a new rating curve.
  A new rating curve is required when part or all of the historic stage-discharge ob-
  servations does not fit new discharge measurements and cannot easily be accom-
  modated by historical rating curve manipulations. Large changes to a water body
  or structural influences on local hydraulics may warrant this reconstruction. An-
  other example would be the construction of a rating curve beyond the maximum
  observed stage-discharge using various types of modeling techniques or a change
  of rating curve from linear table to logarithmic table.
- **Shift:** The shift of a rating curve happens when the entire or part of the rating
  curve needs to be adjusted based on new discharge measurements (but not entirely
  reconstructed). These shifts can have various forms; the simplest form is a con-
  stant or single point shift in which the new observational points show a single value
  shift in comparison to earlier observations and the rating curve (constant over the
  range of the rating curve). The other types of shift can be used to accommodate
  part of the rating curve shift, called knee bend, or more local accommodation of
  changes in the rating curve by truss shift (Figure-3). Readers are encouraged to
  refer to earlier works to read a more extensive elaboration of rating curve shift (Rainville
  et al., 2002; Mansanarez et al., 2019; Reitan & Petersen-Øverleir, 2011).
- **Temporary shift:** The concept of the temporary shift of rating curves is not widely
  known or explored in the literature. The temporary shift is the movement of a rat-
  ing curve along its stage axis to adjust for the short-term presence of environmen-
  tal disturbances such as backwater and ice conditions. Figure-4a-c shows an ex-
  ample of how the temporary shift is applied over time and how the application of





Table 2: Rating curve and discharge estimation definitions

| Definition | Description |
| --- | --- |
| Rating curve | Rating curve is a function that relates an observed stage expressed in the unit of meters [or length] to discharge in volume per time such as cubic meter per second [or volume per time]. A rating curve and its rating curve points are decided by hydrographers based on various factors and past discharge activities (refer to Figure-2). |
| Rating curve points | Rating curve points are the points that define the rating curve functions. The function between the rating points is defined in two ways based on rating curve types. |
| Observational or gauging points | Stage and discharge pair of values that are collected/measured during discharge activity and are used for rating curve creation or temporary shift and override estimation. |
| Rating curve tables or types | The type of functions between the rating curve points. Water Survey of Canada uses either linear or logarithmic tables to define the form of function between consecutive rating curve points |
| Linear Table | Linear relationship is assumed between the two consecutive rating curve points |
| Logarithmic Table | Logarithmic relationship is assumed between the consecutive curve points that follow formulation in form of $Q_t = a(H_t - O)^b$ in which $O$ is the offset (similar to intercept) and is archived in the operational database while $a$, $b$ must be inferred based on the provided starting and ending points of the logarithmic rating curve segment. $H_t$ is the measured stage and $Q_t$ is estimated discharge for time $t$ |
| Offset | Offset identifies the logarithmic function between the two consecutive rating points and accompanies the rating points information in the operational database. The two consecutive rating points and offset are needed to calculate $a$ and $b$ parameters for logarithmic tables. |
| Rating curve shift | Rating curve shifts are permanent shifts of entire or parts of the rating curve to accommodate the systematic changes of observational or gauging points over time |
| Rating curve temporary shift | Rating curve temporary shifts are the time-dependent values in units of length such as meters that the rating curve is shifted for (hence an identical stage value and rating curve result in different discharge given different shift values). Temporary shift values are assigned on a specified date. The temporary shift is then assumed to linearly change between the temporary shift values at two consecutive dates of temporary shift application. |
| Override | Override is a process of correcting the discharge values. Override will result in discharge values being different from what is calculated using stage values, rating curves, and temporary shift values. |

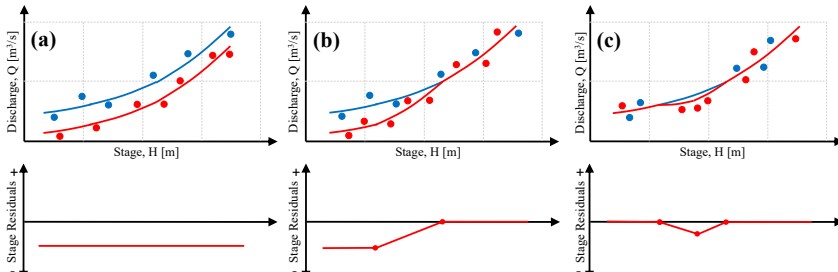

Figure 3: The shift of rating curve segments to accommodate new observation points based on stage residuals for various types from a base [original] rating curve: (a) constant or single point shift in which the rating curve is shifted with a constant value over its entire range, (b) knee bend in which part of rating curve is shifted with a constant value, and (c) truss in which more local shift is applied on a rating curve.

temporary shift affects the inferred discharge compared to the case when no tem-
porary shift is used for ice cover condition. Figure-5 illustrated the effect of ap-
plied temporary shift on the rating curve. Initially, the temporary shift is set to
zero before the time $t_1$ meaning that the stage-discharge relationship follows the
original rating curve. There is a field measurement during this period. The newly
obtained stage and discharge values during the field measurement do not conform
with the rating curve (residuals are not zero). In the next discharge activity dur-
ing the freeze-up period, the hydrographer, based on environmental conditions and
discharge activity at $t_2$, will apply a negative shift. The negative shift can be ei-
ther summed with stage values or can be represented by a rating curve temporary
shift to the positive stage direction (and another way around for positive tempo-
rary shift values). In this example, the rating curve is shifted to the right along
the stage axis, which implies that during the freezing-up period, identical stage
values will result in a smaller discharge estimation in comparison to the original
rating curve (when the temporary shift of zero - open water). The magnitude of
this negative shift is applied as such so that the observed stage and discharge at
time $t_2$ coincides with the temporarily shifted rating curve (observation is given
more weight which results in zero residuals). The temporary shift magnitude is
increased at time $t_3$ based on the development of ice cover over the river. At the
time $t_4$ another discharge activity is performed. The hydrographer decides to ad-
just the temporary shift value at this time, $t_4$, to match the observational stage
and discharge (again giving more weight to observation and setting the residuals
to be minimum). And finally, during a field visit after the ice breaks up, the hy-
drographer reduces the shift magnitude to be set to zero at $t_6$ after which the orig-
inal rating curve is used. The temporary shift changes linearly between the date
and time of application of each temporary shift value. This linear change over time
essentially means that between times of $t_1$ and $t_6$ there is effectively a new rat-
ing curve for every logger reading of stage values. The temporary shift values and
their time and date of application are recorded in the operational database.

## 2.5  Overrides

In addition to the temporary shift of the rating curve, WSC uses other methods
outside the manipulation of rating curves to report an updated discharge estimation. These

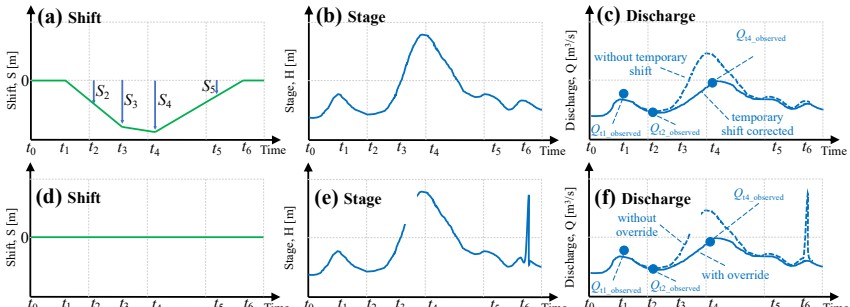

Figure 4: Above panels provide an example of discharge estimation using the concept of temporary shift. The bottom panels provide an example of discharge estimation using the concept of override (while temporary shift is set to zero). (a) The evolution of temporary shifts over time, (b) measured stage time series, (c) estimated discharge time series with and without temporary shift, (d) temporary shift time series, set to zero, (e) stage values record that has a gap and faulty reading, and (f) the estimated discharge values using override techniques that are corrected for the gap, discharge activity, and faulty reading. The effect of temporary shift time series on the rating curve is illustrated in Figure-5

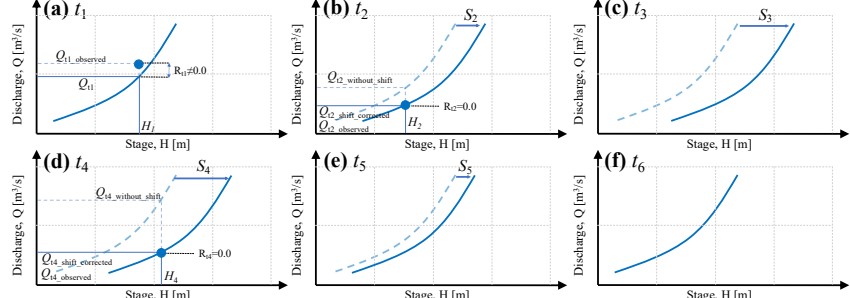

Figure 5: Temporary shifted rating curves at (a) $t_1$, (b) $t_2$, (c) $t_3$, (d) $t_4$, (e) $t_5$, (f) $t_6$ from shift time series illustrated in Figure-4-a applied based on the environmental condition during ice over, hydrographer experience and discharge activities.



updates follow WSC SOP rules and are based on a multitude of factors such as discharge measurements, and the hydrographer's judgment as to the state of changes in the river. The collective title of these efforts is *override* in which WSC hydrographers use various techniques and sources of information to manually correct discharge values. Overrides may include adjustments based on upstream or downstream station readings, linear interpolation of missing values, reconstruction of peak discharge by [hydraulic] modeling, falling limp using decay functions, or under-ice discharge variations among others. The override practices can sometimes vary between the WSC offices. Although the hydrographers at WSC follow SOP guidelines and their experience for this estimation, given that our efforts were limited to data available from the API, it is challenging to easily recreate estimated discharge values reported in the operational database. Figure-4d-f illustrates a very simplified example of an override in which the temporary shift is not used (and hence zero). The discharge values are manipulated to fill the gap between time $t_3$ and $t_4$ in the stage record for the rising limb of a flood event. The discharge values are also changed to reduce the estimated peak flow to better match the observational discharge at time $t_4$. Finally, the hydrographer decides that the stage reading values at $t_6$ are faulty and should not be used for discharge estimation. The discharge values for this faulty reading are then interpolated using the past and future readings of this station and possible existing upstream and/or downstream stations.

### 2.6 Developing an independent Workflow

An independent Python workflow is designed to evaluate the reported discharge values in the operational WSC database. The designed workflow uses the application programming interface or API to extract data directly from the database. The main aim of the workflow is to replicate the reported discharge in the operational database, *Discharge.Historical.Working*, using the recorder stage values, identified by *Stage.Historical.Working*, and other available information, such as rating curves, and temporary shift from the operational database. The workflow is designed into five steps: step-1 is the interrogation of the metadata from the production database. This includes downloading the metadata for available time series at logger resolution such as stage, and other parameters such as pressure, voltage, or any parameter that reflect on the functionality of instruments or environmental factors. Information about the rating curves (their IDs) and the dates of their applications are also extracted. In the second step, step-2, rating curves, and time series are downloaded from the production database. These data are the rating curve tables, including the offset for the logarithmic table, and the effective shift at a given date and time (specified in the shift metadata, from step-1). Step-3 is the adjustment of the variables to common scales. This includes refining the rating curves to increments of 1 millimeter for finer interpolation along the stage axis and also re-sampling, interpolating continuous or discrete information such as temporary shift values, and rating curves ID to temporal stage resolutions. This step provides the needed information for estimating the discharge from stage values. Step-4 mainly focuses on estimating discharge from the stage based on the files created from the adjustment step and the time series of stage values used to recreate discharge within the production system. Finally, step-5 of the workflow focuses on evaluating and interpreting the reproduced discharge and comparison with reported values from the production database. The difference between the reported discharge values in the production database, which includes override practices and values, and reconstructed discharge based on the above-mentioned workflow can shed light on the level of possible intervention by override or other methods on reported discharge.





## 3 Results

### 3.1 Rating Curves Construction and Characteristics

Rating curves are characterized by rating points, and in the case of a logarithmic table, they are accompanied by offset values ($O$, refer to Table-2 and Figure-2). Our findings, contrasting the rating curves and observational points, indicate that the creation of rating curves from observational points does not always follow a unified statistical approach. Rather, it is sometimes based on hydrographers' judgment and field observations. Additionally, it is not apparent, when extracting data from the API system, which stage-discharge measurement points are used to update the current rating. A few of the limitations in reproducing rating curves are described below. (Figure-6):

- **Rating curve extrapolation/extension beyond the largest stage-discharge in the operational database record:** The rating curves might be extended beyond the largest stage and discharge observed values in the operational database. The method for the extension of the rating curves is not provided through the API in the operational database. Very old observational points that are not recorded in the operational database may be used in creating more recent rating curves or the extrapolation is done using hydraulic modeling or other procedures. For example, the difference in the rating curves for station 02YR004 is perhaps due to extrapolation outside the range of maximum observation using SOPs. For earlier rating curves that use linear tables this extrapolation is linear while for more recent rating curves expressed in the logarithmic table, the extrapolation is done in logarithmic space. (Figure-6a).
- **Extrapolation of rating curve for out-of-bank conditions:** one of the difficulties is to construct the rating curve for the out-of-bank condition with limited observational points at high water conditions (Figure-6b).
- **Removal of ice-conditioned stage-discharge points:** The formation of an ice cover causes increased friction and generates a backwater effect where the water level has a different relationship to discharge than in open water conditions. Under ice observational points have much lower river discharge in comparison to open water flow for the same stage values and therefore are not used in the construction of rating curves, instead are used to adjust the estimated discharge using override values or temporary shifts during the ice condition (Figure-6c). This, in turn, results in fewer observational points being available for the construction of rating curves.
- **Emphasis on one observational point:** A rating curve is often created or changed based on one gauging measurement. Observational points with very high discharge values can affect the higher end of the rating curve. This can be due to high discharge values only occurring for brief periods resulting in one observation in the high discharge period being the only observation. In the example provided for station 01FF001, an observational point with stage and discharge of approximately 1.75 m and 40 m3/s is given very high weight in creating the immediate rating curve update after the aforementioned field activity while in later rating curves, this high emphasis is not followed (Figure-6d).
- **Event-based erosion, flood, or long-term channel erosion:** River section may change over time and therefore observational stage and discharge points follow these changes accordingly. Sediment transport occurs gradually and over longer periods than a flood event, but can result in complex changes in the measurement section as sediment is deposited or removed or as dunes proceed through the section. These changes require a new rating curve or shifts in the existing rating curve (Figure-6e). Similarly, floods or high water levels can also result in a substantial change in river section or removal of stations. In these cases, a new rating curve is needed.



- **Changes in rating curve benchmark stage or instrument stage reading change:** A benchmark is a fixed point that is used to link the observed water level to an actual elevation. The local benchmark that is used as a datum may change over time with the landscape or administrative change. Alternately instrument replacement, after a flood event for example, in a new location can also change the reading in comparison to historical readings compared to the benchmark (Figure-6f).

Given the above, it is important to emphasize that the use of rating curves within the Water Survey of Canada does not allow for a more classic statistical approach for uncertainty analysis where the curve would be the best fit through the series of observed points (as it is for other institutions such as UK environmental agency Coxon et al., 2015). The actual process used is deterministic and much effort is invested in making the rating curve pass through or close to each measurement, or stage and discharge point, which has been a long-standing practical approach (Rantz, 1982). This, however, means that the residual structure may not follow a known statistical model, may change from location to location, and is subjected to hydrographers' experience and judgment. This is elaborated further in the following subsection about the structure of residuals. Observed stage-discharge records are not random samples since they have a time sequence and a measurement bias. For example, high discharges only occur for brief periods and are less frequent than lower discharges. Conducting discharge activities might be dangerous and challenging during high water, and many rivers in a region peak simultaneously in time, so there is a systematic under-representation of high discharge values. This lack of stage-discharge observations might be particularly important for the stations that are located on sections that are not stable (Whitfield & Pomeroy, 2017).

Seasonality and ice condition are other factors that can complicate the use of existing stage-discharge observations. When there is ice cover, the stage-discharge relationship will vary substantially from the expected open-water rating curves. Figure-7 indicated that the stage-discharge measurements during cold months of the year were identified by flag B, or backwater due to ice, in contrast to those without any or other flags. As it is clear from panels of Figure-7, the winter period often has smaller discharge values for a similar stage to those in summer, therefore, resulting in a smaller pool of stage-discharge observation that could be used for rating curve creation.

Additionally, Figure-8 provides fractions of discharge activities, discharge values, and ice flags for each specific month of the year for the entire hydrometric network and 11 major drainage basins in Canada. The red dashed line indicates the change over the year for the percent of each month's field discharge measurements from total discharge measurements while the blue line provides an understanding of the magnitude of the discharge values over the month of a year. The shaded blue for each month provides the comparison between the fraction of time that the stations times series for that month are identified by flag B (which is used to identify backwaters due to ice conditions). The number of discharge field measurement activities during the summer months is larger than in the winter months. This is due to the spring and summer variability in discharge being much greater than in winter and because ice discharge measurements are expensive and labor-intensive in comparison to open-water measurements.

Evaluating the recorded stage greater than the maximum observed stage in the operational database provides an understanding of how often discharge estimates are in the portions of extrapolated rating curves beyond the observed stage-discharge points that are archived in the operational database. Figure-9 indicates that there are stations in which the stage higher than the maximum observed stage during discharge activity can occur in any month of the year. One example of this is 02YR004; Triton Brook above Gambo Pond in the province of Newfoundland and Labrador (Figure-6a). This could happen because the operational database might not include earlier stage-discharge measurements with the highest stage values or systematic backwater from increased water level in Gambo pond. In general, Figure-9 highlights the existence of numerous events when

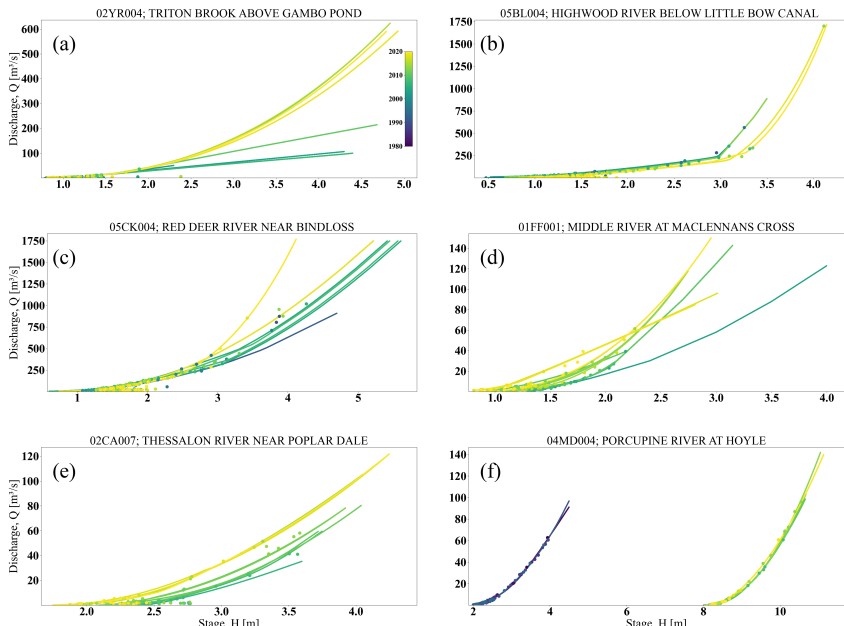

Figure 6: Example of rating curves and observations available in Aquarius illustrating rating curves over time where (a) curves are extended outside of the highest discharge observation extrapolation (b) sharp breaks in rating curves when the river flows out of bank (c) under ice stage-discharge observations are not used in rating curve creation, (d) emphasis on one point of observation results in a change to the rating curve, (e) long or short term river bed erosion, and (f) change in rating curve benchmark for reporting stage values.



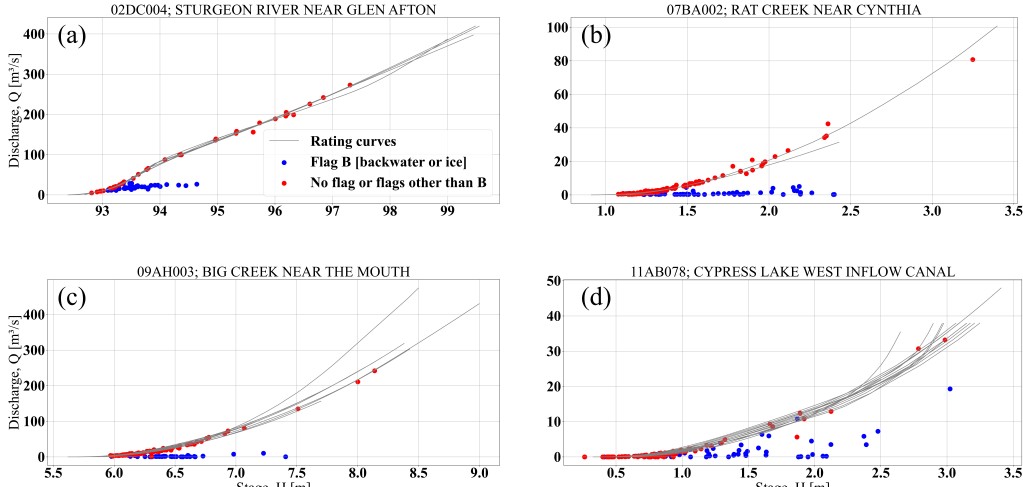

Figure 7: The contrast between the stage-discharge measurements with and without the B flag for stations (a) 2DC004, Sturgeon River Near Glen Afton, (b) 07BA002, Rat Creek Near Cynthia, (c) 09AH003, Big Creek Near The Mouth, and (d) 11AB078, Cypress Lake West Inflow Canal. The red points do not have flags while the blue points are stage-discharge measurements that have the B flag, ice or backwater, in the operational database.

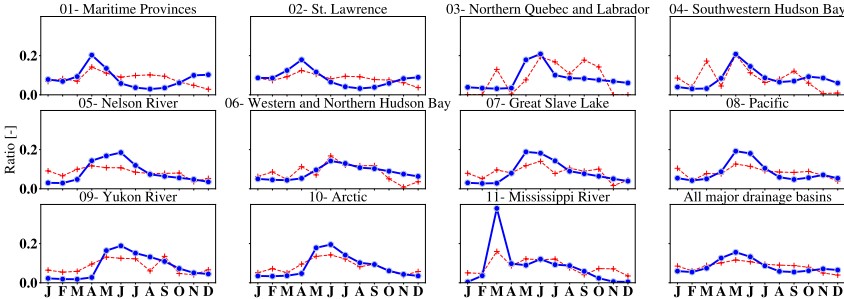

Figure 8: The blue and red dotted lines indicate the fraction of annual discharge and of annual discharge activity respectively, for each major drainage basin and for all drainage basins (the total of existing stations in the WSC operational database). The blue shading identifies the fraction of time series that are identified by flag B or backwater that is used to identify ice conditions. The darker the shade the more dominant flag B or ice cover is for the major drainage basin.



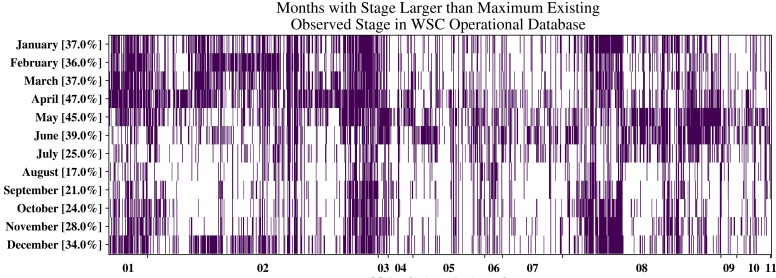

Figure 9: Months where the recorded stage values exceed the maximum observed stage during any discharge activities archived in the operational database. A solid bar in the month in the figure indicates, for a station and during its available record, there is at least one event in that month across all years, with recorded stage values exceeding the maximum observed stage value. The percentage for each month indicates the fraction of stations where the recorded stage is exceeding the maximum observed stage and discharge.

discharge values are estimated using extrapolated segments which can have significant impacts on estimates of discharge and its uncertainty in flood modeling and flood forecasting.

The temporary shift of rating curves to account for environmental conditions is a common practice at the regional offices of WSC. Figure-10 identified three major characteristics of temporary shift application across the Canadian hydrometric stations. First is the average number of days per year in which temporary shift is applied (Figure-10a). For the prairie regions, especially stations operated by the Calgary office in the province of Alberta, the temporary shift can be applied all year long (length of temporary shift application larger than 300 days per year). As presented in Figure-10, using the temporary shift to adjust for environmental conditions is most common in Prairie and Northern regions. The use of temporary shifts is less common in Eastern and Western Canada. In those regions, direct manipulation of discharge values rather than the rating curves is more common (following override). The second panel, Figure-10b, indicates the magnitude of temporary shift applied in meters. There are stations with temporary shift magnitude of more than 1 meter; this means during various environmental conditions such as the presence of thick ice cover, stage values that are as different as one meter or more, under the temporary shift application, may result in similar discharge estimation. Lastly, Figure-10c, identified the range of applied temporary shift to the range of stage values. This comparison indicates how relative intervention by temporary shift is compared to the changes in recorded stage values. Interestingly, there are stations over the Canadian domain in which the range of temporary shift surpass the range of recorded stage values (ratio of more than one).

## 3.2 Time series reconstruction

In steps 3 & 4 of the independent workflow, river discharge values are reconstructed and compared with the reported discharge values from the WSC operational database. This comparison of discharge values indicates four categories for discharge estimation:

1. **Rating curve:** in which the estimated discharge values strictly follow the stage-discharge relationship or rating curves and can be reconstructed using stage values.

Figure 10: (a) Temporal application of temporary shift, (b) range of applied temporary shift, and (c) the ratio of temporary shift range to stage range across hydrometric stations of Water Survey of Canada. The orange and red colors in the background indicate the major drainage basins (refer to Figure-1).



2. **Temporary shift:** in which the discharge follows the temporarily shifted rating curves and can be reconstructed using stage values.

3. **Override:** The period in which the discharge is estimated using override methods and techniques (not following rating curve and temporary shift).

4. **Temporary shift and override:** in which both temporary shift of rating curve and override methods are applied at the same time.

Table-3 indicates the four categories of discharge estimation, and their reproducibility using the independent Python workflow, given the data that was retrievable from the API system.

To provide clear examples of each of the categories, four stations are examined. Figure-11 illustrates the recorded stage for 01AF009, Iroquois River at Moulin Morneault located in the province of New Brunswick, in the top panel, the applied shift, and the date of field or discharge activities shown in the second panel from the top. The third panel from the top compares the recreated discharge, using the workflow described in this study, and the reported discharge from the operational database. The shaded areas in this panel indicate the quality assessment symbol (flag) from the operational dataset. The temporary shift values applied for the year 2003 are zero. However, the under-ice condition in the reported discharge values from the operational database is significantly lower than the reconstructed discharge values from the stage using the rating curves and temporary shift of zero values. The under-ice discharge estimate is an override applied using various methods at the regional offices. It can be seen that override discharge values pass through the observational points under ice conditions, these observations of discharge are the basis for the winter flow record and not the recorded stage and the rating curve, while the variation is also recreated following established logic at the regional office such as under ice peak flows (in this example, late March and early April). This is reflected in the bottom panel in which two major discharge estimation categories are depicted: the green is when rating curves are followed without temporary shift and the gold is when the override methods are applied.

Discharge values for station 05BL004; Highwood River Below Little Bow Canal is provided in Figure-12. The hydrographers have applied negative temporary shifts for this station. For the year 2012, the temporary shift was applied during winter with larger shifts (-0.25 to -0.50) and during summer with rather small shifts (<-0.20). The winter shift is presumed to be correcting for ice conditions and the summer shift, in June, is likely for the backwater correction over the high discharge period (while there is no associated flag with this event). Temporary shifts are sometimes applied on dates that coincide with discharge activities or site visits, presumably to match the observed discharge with the rating curve with temporary shifts. Shift values can be changed on other dates that might correspond with temperature changes or video recordings from on-site monitoring cameras or upstream and downstream station field visits and observations. The bottom panel indicated that for this station and the year of interest, there are two major discharge estimation categories: the blue is the rating curve and temporary shift and the magenta is rating curve and temporary shift which is corrected by override (slightly in this case).

Discharge values for station 08GA079; Seymour River Above Lakehead is given in Figure-13. There is no application of temporary shift and override for this station in the year 2002 and therefore estimated discharge follows the rating curve concept (presented by green in the bottom panel).

The last example focuses on station 09CB001; White River at Kilometer 1881.6 Alaska Highway in Yukon Territory (Figure-14). This is an example of a station in which a variety of discharge estimation methods are used. In part of summer, the discharge can be fully reproduced by rating curves. There are also periods that the temporary shift is applied over summer and discharge estimation follows the rating curve and temporary shift. In part of the summer, in addition to the temporary shift concept, the override is also applied to correct the estimated discharge. For the winter period, there is no applica-





Table 3: Types of discharge estimation

| Discharge estimation categories | Condition of application | Reproducibility | Uncertainty |
|---|---|---|---|
| Rating curve | Open water condition. Environmental conditions are not significant enough to result in deviation from the stage-discharge relationship or rating curve. | Fully reproducible discharge values following the stage and rating curve. | The discharge uncertainty estimation can be attributed to rating curve uncertainty (type A). |
| Temporary shift | Backwater, under ice conditions, temporarily changes to the channel. The rating curve is temporarily adjusted to accommodate environmental conditions affecting the stage-discharge relationship. | Fully reproducible discharge values following the stage, temporary shift, and rating curve. However, the magnitude of shift values and their time of applications are based on hydrographer judgment and may not be easily reproducible. | Often a magnitude of the temporary shift is applied, resulting in the highest agreement between observed discharge and estimated discharge (using temporary shift). The residuals are therefore suppressed to small values. Uncertainty estimation methods should be sought to handle the uncertainty estimation of temporary shift practice, type B, in addition to the rating curve uncertainty, type A, resulting in a composite uncertainty model (type A+B) |
| Override | Stable backwater or under ice conditions, correction of the erroneous values, gap filling of missing data, estimation of freeze up or ice break up transition or ice jams. | Not reproducible following the stage and rating-curve concept; Greatly reproducible using the Aquarius™ and available techniques, trained WSC hydrographers. | Estimation of discharge using override gives higher weight to discharge observation that suppresses the residuals (similar to temporary shift). The various methods that are used for override may have various levels of uncertainties which are also dependent on the hydrographers' skills. New uncertainty methods are needed to account for these complexities (type C). |
| Temporary shift and override (mixed) | All the conditions for temporary shift and override. In this case, the discharge is estimated using a temporary shift and override simultaneously to correct the discharge values further. | Not reproducible following the stage and rating-curve concept. Greatly reproducible using the Aquarius™ and available techniques, trained WSC hydrographers. | The challenges of uncertainty estimation under temporary shift and override can be addressed by developing uncertainty methods for override and temporary shift (type A+B+C). |

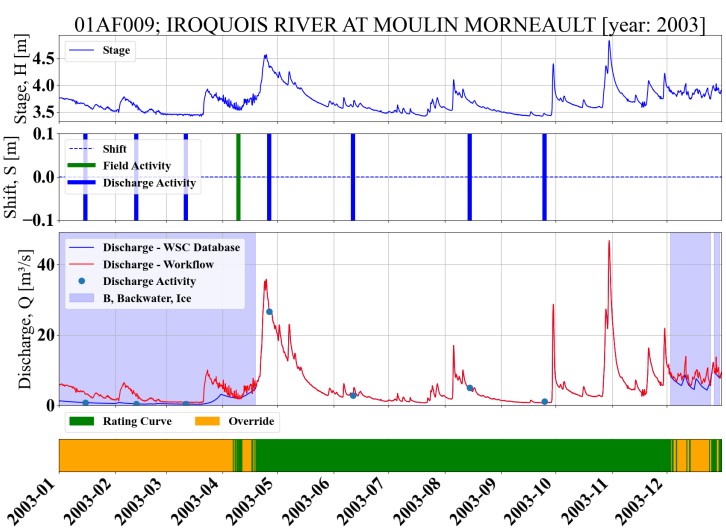

Figure 11: (Top panel) the recorded stage, (second panel from top) the applied temporary shift, (third panel from top) reproduced discharge values based on workflow and comparison to reported discharge values from operational database and discharge activities, and (bottom panel) dominated method of discharge estimation for 01AF009; Iroquois River at Moulin Morneault located in the province of New Brunswick. The colors in the lower bar link to the descriptions in Table-3: rating curve (green), and override (gold).

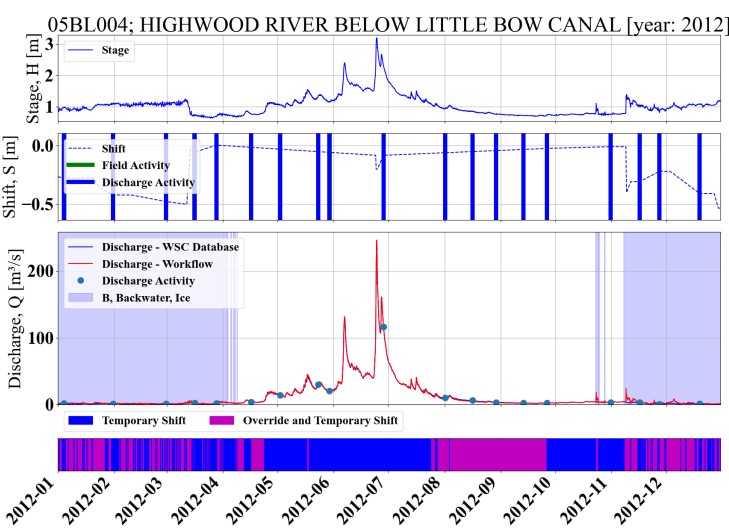

Figure 12: (Top panel) the recorded stage, (second panel from top) the applied temporary shift, (third panel from top) reproduced discharge values based on workflow and comparison to reported discharge values from operational database and discharge activities, and (bottom panel) dominated method of discharge estimation for 05BL004; Highwood River Below Little Bow Canal located in the province of Alberta. The colors in the lower bar link to the descriptions in Table-3: temporary shift (blue), override with temporary shift, and override (magenta).



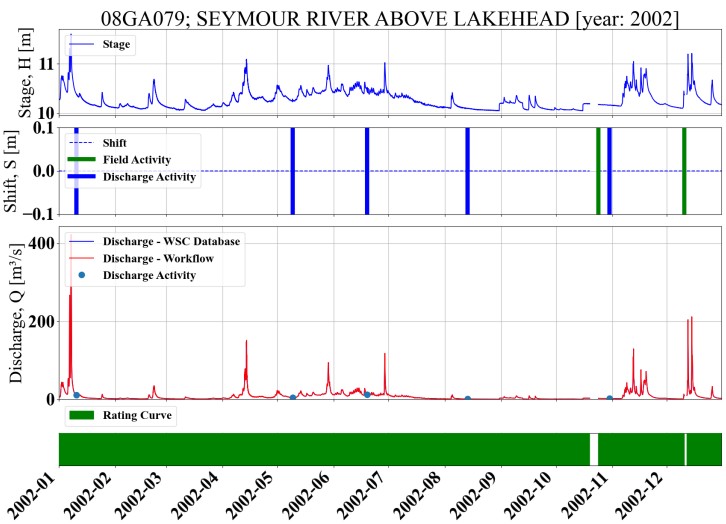

Figure 13: (Top panel) the recorded stage, (second panel from top) the applied temporary shift, (third panel from top) reproduced discharge values based on workflow and comparison to reported discharge values from operational database and discharge activities, and (bottom panel) dominated method of discharge estimation for 08GA079; Seymour River Above Lakehead in the province of British Columbia. The colors in the lower bar link to the descriptions in Table-3: rating curve (green), infilled or missing data (white).

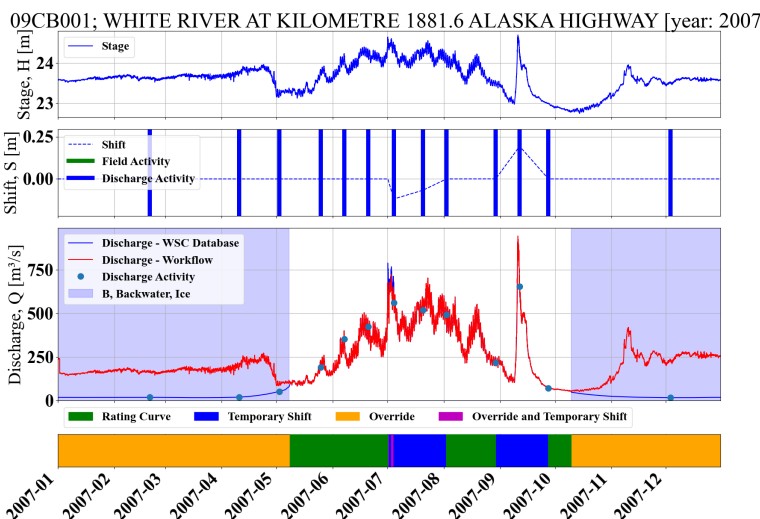

Figure 14: (Top panel) the recorded stage, (second panel from top) the applied temporary shift, (third panel from top) reproduced discharge values based on workflow and comparison to reported discharge values from operational database and discharge activities, and (bottom panel) dominated method of discharge estimation for 09CB001; White River at Kilometer 1881.6 Alaska Highway in Yukon Territory. The colors in the lower bar link to the descriptions in Table-3: rating curve (green), override (gold), temporary shift (blue), and, override with temporary shift and override (magenta).

tion of temporary shift, however, the override is used by emphasizing the observation, perhaps under ice observation, to estimate discharge (similar to Figure-12).

Given the difference between the reproduced and reported discharge values in the operational database, similar to stations 01AF009, in the following, the agreement between the reported discharge in the operational database was evaluated using the independent workflow for all the hydrometric stations that have a complete yearly record (not seasonal). Figure-15 depicts this agreement in a fraction of the period in which reconstructed discharge is within 5% of the discharge reported in the operational database. The overall overlap is around 0.67. This level of agreement from the independent workflow can be attributed to discharge estimation from rating curves and rating curves combined with the temporary shift. On the other hand, the lack of agreement can be heavily attributed to the override values which are more pronounced during the winter period. This lack of agreement can be also partly attributed to the types of data that are not available from the WSC operational database via the API (that is used for the workflow). Trained and experienced WSC hydrographers can reproduce discharge values, with great similarities if not identical, using the Aquarius™, documented comments in the operational database. This is also checked and confirmed during the approval process. Therefore the reproducibility, in practice, will be much higher than the general agreement which is stated here. As an example, if the discharge values under ice are given higher prior-



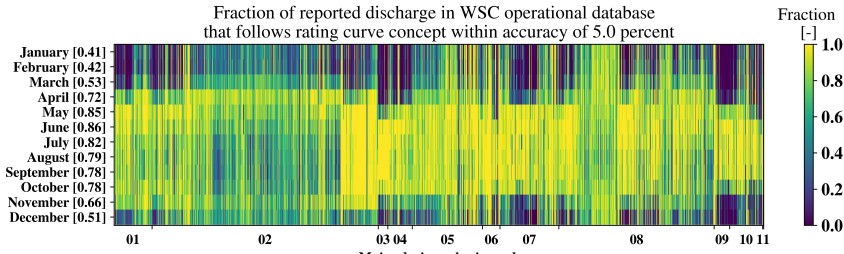

Figure 15: The fraction of agreement for estimated discharge values from the proposed workflow described in this study (within 5% of reported discharge values from the WSC operational database). The agreement fraction is not always to its maximum, 1.00, and varies seasonally and geographically. The overall average agreement between the recreated discharge values and what is reported in the operational database is 0.67, with winter months having lower agreement than the summer months.

ity and the discharge for the ice cover period is interpolated using a linear interpolation technique the overall reported agreement from the workflow to reported discharge values of the operational database increases to 74% (from 67%).

### 3.2.1 Implication for Uncertainty Estimation

The procedures and practices at WSC, namely override and temporary shift, will result in different residual structures than those often expected to represent the structure of residuals in the literature. Figure-11 to 14, indicate that observational stage-discharge measurements are weighted heavily in discharge estimation. To investigate, the reported discharge values from the WSC operational database, which includes override and shift, in pair with observational discharge are compared with the case of Gaussian distribution with heteroscedastic errors. Figure-16 illustrates this contrast for four stations (01AJ004, 04AB001, 05AA008, and 07AH003). The reported discharge in the operational database matches the measured discharge (very close to the line of perfect agreement) while the structure of the expected residuals, represented as grey points, is far more scattered. This hints at deficiencies of existing models for residual estimation, assuming that the observations are without error, across the Canadian hydrometric stations due to override and temporary shift among other SOPs.

A closer examination of the interaction of the stage and reported discharge values to observational points depicts two relationships for each of the stations mentioned in Figure-16. In Figure-17, the right panels indicate the rating curves while the left panels depict the time-series relationship between all reported stage and discharge values from the WSC operational database, which include temporary shifts and overrides, in contrast to observational stage-discharge points. Comparing the right and left panels indicates that the stage-discharge relationships or rating curves may not incorporate stage-discharge observation points while the stage-discharge space, left panels, conform with observational stage-discharge. This highlights to some degree why shifts and overrides need to be applied since the classical curve fitting technique to all available observational stage-discharge points would not reflect the local hydraulic realities at the time of measurement. The observational points have a much more complicated relationship with the rating curves than standard curve fitting practice (Figure-17).

High Flows are critical data points in annual maxima time-series analysis. The flood of June 2013 for station 05AA035, Oldman River at Range Road No. 13A, Alberta, is selected to assess both discharge estimation practices and implications for uncertainty

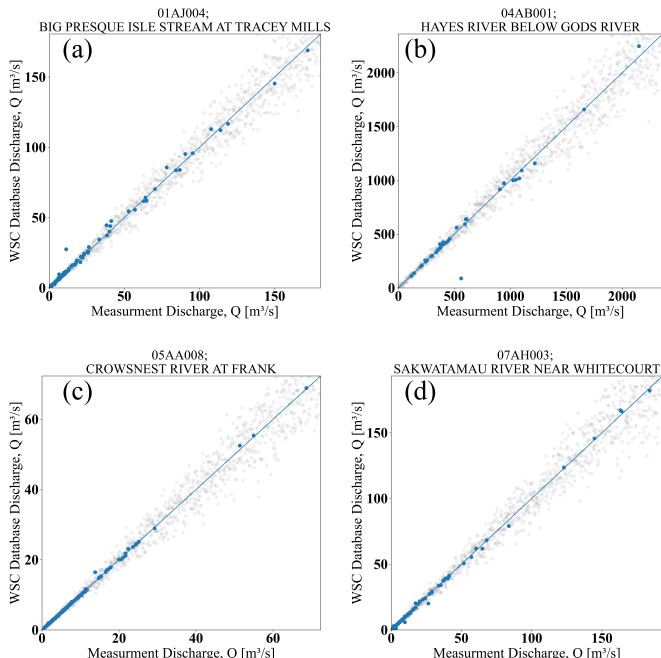

Figure 16: The comparison between discharge values reported in the WSC operational database at logger resolution and measured discharge during discharge activity in blue dots, for stations (a) 01AJ004; Big Presque Isle Stream at Tracey Mills, New Brunswick, (b) 04AB001; Hayes River Below Gods River, Manitoba, (c) 05AA008; Crowsnest River at Frank, Alberta, and (d) 07AH003; Sakwatamau River Near Whitecourt, Alberta. In contrast, the gray dots are the hypothetical case of the normal distribution with a heteroscedastic standard deviation of 10% of discharge magnitude. The blue line, 1:1, is the best-expected fit for these two series.

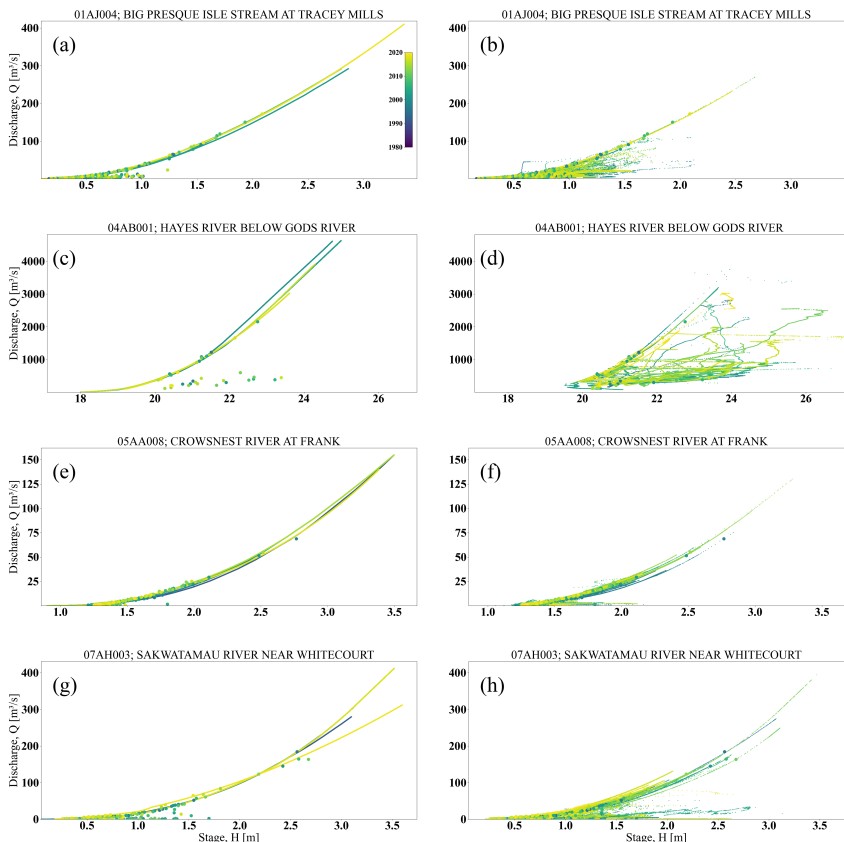

Figure 17: The comparison of stage-discharge rating curves (left panels) and observed stage and reported discharge and stage values from the WSC operational database (right panels) contrasting observational stage-discharge points obtained during discharge activities for stations (a,b) 01AJ004; Big Presque Isle Stream at Tracey Mills, New Brunswick, (c,d) 04AB001; Hayes River Below Gods River, Manitoba, (e,f) 05AA008; Crowsnest River at Frank, Alberta, and (g,h) 07AH003; Sakwatamau River Near Whitecourt, Alberta.

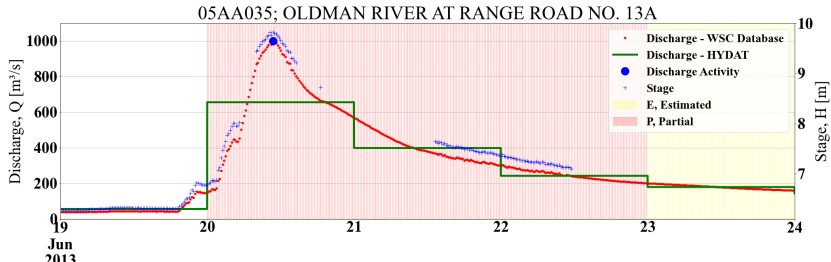

Figure 18: The comparison between the reported discharge and stage values at logger temporal resolution from the operational database, measured discharge at the flood peak, and HYDAT reported daily discharge and flags for Station 05AA035, Oldman River at Range Road No. 13A, Alberta.

analysis. The comparison presented in Figure-18 indicates that the reported discharge values from the operational database are as high as 1000 cubic meters per second and conform with the stage-discharge measurement at approximately 10:00 AM local time (residual of zero). Stage values are not continuously measured at 5-minute intervals during the flood period (Figure-18). This result in the flag "P" *partial* being applied; there is only a partial stage available for days for $20^{th}$, $21^{th}$, and $22^{th}$ June. The estimated/filled discharge values at logger resolution are smoothed, and there is less variation, while for the time when the stage is available, discharge exhibits more variation given the variability in the stage. The stage values are fully missing for $23^{th}$ June and therefore the entire discharge values for that day are identified with the flag "E" *estimated*. The override metadata file, extracted from the operational database, reports that the gap filling during this period is performed using meteorological information, comparison with other stations, and linear approximation under the general procedure of *multi-points drift correction* at the regional office (but does not provide quantitative values for this approximation). In general, it should be noted that the sub-daily variability which can be significantly important is lost due to this temporal aggregation, and the instantaneous maximum yearly flow communicated in the HYDAT dataset may not be sufficient to reconstruct sub-daily variability or residuals. The reported daily values for $20^{th}$ of June 2013 is 655A $m^3/s$ which is 345 $m^3/s$ lower than the measured discharge in the field and also what the operational database reports. Care should be taken when using daily discharge values for modeling and decision-making, and residual evaluation for uncertainty estimation.

Given the WSC SOPs on residuals, each discharge estimation category mentioned in Table-3 should have its suitable discharge uncertainty models. For example, when the rating curve is used for discharge estimation, rating curve uncertainty, which has been heavily studied in the literature, can be used (type A from Table-3). However, WSC hydrometric stations do require a more tailored method than what is often suggested in the literature due to temporary shift and override as part of SOPs. When the temporary shift concept is followed, a new method, in which both the rating curve and temporary shift uncertainty are estimated is needed and an uncertainty model to account for temporary shifts needs to be formulated, type B, in addition to rating curve uncertainty, type A. The discharge uncertainty would then be the interaction of the two models (type A+B). This becomes even more challenging when the override is used for discharge estimation; more sophisticated uncertainty estimation techniques may be essential to be developed (type C). Additionally, the fact that the discharge estimation technique may change throughout each season adds to this complexity as well (translation





between uncertainty models across time). Furthermore, reproducibility can be seen as
the cornerstone of the uncertainty models. For example, to be able to create a model for
uncertainty type C, perhaps a discharge estimation model with associated parameters
should be formulated during override periods. The discharge estimation model then can
be used for perturbation and uncertainty analysis (similar to uncertainty estimation of
rating curves, type A).

Finally, a simple experiment is designed to generate an ensemble of discharge estimations for evaluating the impact of decisions such as rating curve creations, temporary shift application, and override, on estimated discharge. For this analysis, stations
are selected for which changes in rating curves over time cannot be differentiated from
observational stage-discharge points. Two stations, 05BA002; Pipestone River Near Lake
Louise, Alberta, and 03OA012; Luce Brook Below Tinto Pond, Newfoundland and Labrador
are considered for this analysis. The workflow is slightly changed to generate ensemble
discharge values: (1) the rating curves are given equal probability and replace each other
in their effective period of applicability and (2) the discharge estimation is done considering temporary shift and without temporary shift (or temporary shift set to zero). The
ensemble members are then compared to the reported discharge values by commonly used
performance metrics in Earth System modeling (runoff ratio, $E_{RR}$, Root Mean Square
Error, $E_{RMSE}$, Nash-Sutcliffe Efficiency, $E_{NSE}$, and Kling-Gupta Efficiency, $E_{KGE}$ (for
further explanation refer to Appendix A).

The dark blue area in Figure-19a indicates the impact of lack of temporary shift
while reshuffling the rating curves (the effect of choice of rating curve construction and
lack of rating curve manipulation by temporary shift). The dark red area indicates the
effect of temporary shift on inferred discharge time series while reshuffling the rating curves
(the effect of choice of rating curve construction and presence of temporary shift). Figure-
19b illustrates these effects for station 03OA012. Due to the absence of shift values (zero
shift), the dark red and blue areas are coinciding and exhibit similar performance metrics compared to the reported database discharge values (no effect of temporary shift for
this station). The comparison between Figure-19a and b indicate that the impact of rating curve construction is more pronounced for station 05BA002 in comparison to station 03OA012 due to the spread of ensemble members.

The mean performance metrics for the ensembles and also discharge values from
the WSC operational database in comparison to HYDAT values are presented in Table-
4. For the station that temporary shift is not used, 03OA012, the difference between the
shift corrected and not shifted rating curves are identical (as expected). However, the
impact of override, in this case, is much more pronounced, and performance increases
from negative or closer to zero values up to the perfect agreement with HYDAT discharge
values for this station. This drastic change in performance metrics is done by choice of
rating curves and override. In contrast, and for the station where temporary shift practice is applied, such as 05BA002, the inclusion of temporary shift can improve the performance in the scale of $E_{NSE}$ or $E_{KGE}$ while the impact of the choice of rating curve
seems to be more pronounced than the case for station 03OA012 (based on comparison
of Figure-19a and b).

## 4 Discussion and Conclusions

This work presents discharge estimation methods used by the Water Survey of Canada
(WSC) following an independent Python workflow. The study explores the Standard Operation Procedures (SOPs) for creating rating curves, manipulating them over time, and
estimating discharge. The study focuses on two major discharge estimation SOPs, namely
temporary shift, and override. The impact of these SOPs on discharge estimation and
uncertainty evaluation, specifically in terms of residuals, is discussed. By examining the
SOPs and their possible impact on discharge estimation and associated uncertainties,
the study aims to highlight the need for new discharge uncertainty methods.

The relationship between the rating curves and observational stage-discharge measurements is explored. The WSC SOPs differ from more commonly used practices in other

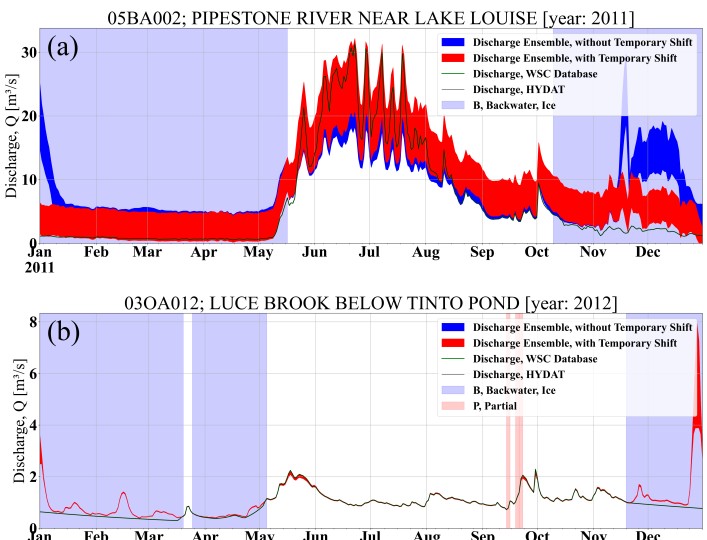

Figure 19: The comparison for the effect of decisions on discharge estimation without shift value, with shift values, reported Aquarius discharge value, and reported HYDAT discharge alongside the flags for (a) 05BA002; Pipestone River Near Lake Louise, Alberta, and (b) 03OA012; Luce Brook Below Tinto Pond, Newfoundland and Labrador.

Table 4: The mean performance of ensemble members with and without shift and discharge values reported in WSC operational database in comparison to HYDAT discharge values.

|  | 05BA002 [year: 2011] | | | | 03OA012 [year: 2012] | | | |
|---|---|---|---|---|---|---|---|---|
|  | $E_{RMSE}$ | $E_{KGE}$ | $E_{NSE}$ | $E_{RR}$ | $E_{RMSE}$ | $E_{KGE}$ | $E_{NSE}$ | $E_{RR}$ |
| without temporary shift | 4.890 | 0.535 | 0.589 | 1.048 | 0.548 | 0.336 | -0.702 | 0.747 |
| with temporary shift | 2.516 | 0.672 | 0.862 | 0.974 | 0.548 | 0.336 | -0.702 | 0.747 |
| WSC operational database | 0.016 | 0.999 | 0.999 | 0.784 | 0.002 | 0.999 | 0.999 | 0.642 |
| HYDAT dataset | 0.000 | 1.000 | 1.000 | 0.785 | 0.000 | 1.000 | 1.000 | 0.642 |





parts of the world (McMillan et al., 2010; Coxon et al., 2015), largely due to the hydrological regimes and conditions faced by the Survey in Canada. Temporary shifts and override processes, while giving the observational stage-discharge a high weight in discharge estimation, resulting in a more complex relationship between the rating curve and observations than a standard curve fitting exercise (Figure-17). This complexity does not lend itself well to more traditional uncertainty approaches. New methods must be explored to evaluate the rating curve uncertainties over and above the already existing methods that rely on the specific nature of residuals, such as heteroscedastic Gaussian, in literature (e.g. methods suggested by Clarke, 1999; Jalbert et al., 2011; Le Coz et al., 2014; Kiang et al., 2018, are not readily applicable for Canadian hydrometric realities).

Following the available information in the WSC operational database accessible by the API and independent Python workflow the agreement level between the two discharge estimations, from the workflow and operational database, is explored. This agreement is significantly lower during the colder months which in turn indicates the complication of the discharge estimation under ice conditions and their backwater effect. To account for this environmental factor, different regional offices may follow different procedures rather than rating curves. In parts of Canada, the override procedure is used, while the Prairie and Northern regions rely heavily on the temporary shift of rating curves (Figure-10).

This study, given the complexity of the production system and updating of rating curve information, encourages the community to consider the provenance of discharge data and evaluate its fitness for its intended use. The discharge values are more than just a true or deterministic value disseminated from the HYDAT dataset. This dataset is often used in large sample hydrology, Gupta et al. (2014), and carried over to the larger datasets without its error and uncertainties being communicated (as an example, Addor et al., 2017; Arsenault et al., 2020; Kratzert et al., 2022, do not carry discharge uncertainty values). These discharge values are then used for scientific purposes, model development, and model inter-comparison alongside recently used machine learning techniques. If uncertainty and errors in discharge are ignored, the use of large sample datasets may result in misleading or strong conclusions. For example, it has been communicated that machine learning can predict the discharge values with 99% percent accuracy or can predict discharge superior to traditionally used mechanistic Earth System models (in literature or blog posts). These comments and conclusions should be taken with care as the hydrographers' decisions in estimating discharge can significantly change a hydrograph (refer to Figure-19 and Table-4). Instead, the efforts should be focused on re-assessing those claims with an ensemble of discharge values. Using an ensemble of discharge timeseries alongside an ensemble of forcing variables of precipitation and temperature can provide a much more robust analysis of scientific methods, decisions, and claims for Earth System models (Cornes et al., 2018; Wong et al., 2021; Tang et al., 2022).

This work provides the basis for future uncertainty analysis of discharge values reported by the Water Survey of Canada. For better estimation of discharge values as an outside user and associated uncertainties, however, more information is needed to be added to the WSC operational database and more capabilities are needed to be developed for Aquarius™ system. This information does exist in WSC offices on paper, field notes, and local computer systems but is not fully transferable to the operational database. As an example, during the preparation of this work and from the API system, it was not possible to find out which observational stage-discharge points are used for rating curve creation. Additionally, the information that might help on observational stage-discharge uncertainty was not available through API to the best of the authors' knowledge. The inclusion of rationale behind the magnitude and date of application of temporary shift or override methods can be a great asset for the operational database. The recommendations transcend the WSC operational procedures and agencies that follow similar approaches to WSC. As an example, The Water Survey of Canada, WSC, and the United State Geological Survey, USGS, have a long history of collaboration going back to the beginning of the WSC mandate in 1908. The chief hydrographer for Canada spent his early years





training with USGS staff in Montana and since then both organizations have developed shared common practices. Both the USGS and WSC use Aquarius™ as their primary data production platform and the practices of overrides and temporary shifts are used by the two organizations. Additional effort is still needed to better access the similarities and implications of procedural practices on discharge estimation and uncertainty quantification between the two countries.

We summarize our major finding as follow:

- The Water Survey of Canada's standard operating procedures in estimating discharge from stage values, particularly temporary shift, and override are explored and explained by an independent Pytho workflow.
- There is no single approach for estimating the rating curve from past observational (stage and discharge) points at the Water Survey of Canada. This is perhaps due to the complex relationship between the stage-discharge relationships accounting for the complexity and diversity of discharge values over the range of environmental conditions for Canadian hydrometric stations. Additionally, given SOPs such as override and temporary shift, relationships between rating curves and observational stage-discharge points are more complex than just a curve-fitting exercise.
- Given the knowledge of discharge estimation processes, the reported discharge values in Aquarius can be reproduced for a fraction of 0.67 (within 5% accuracy). The other 0.33 non-reproducible fraction can be heavily attributed to the override.
- The standard operating procedures, or SOPs, of temporary shift and override result in the residuals being suppressed to minimal values. These will not follow the often assumed statistical distributions for residuals or fundamental basis for rating curve uncertainty estimation methods. Additional uncertainty models for rating curves that do not have structured residuals in comparison to stage and discharge measurements, temporary shift, and override techniques should be constructed and evaluated for Canadian hydrometric stations (uncertainty models of type A, B, and C from Tabel-3).
- Additionally, the impact of SOPs on discharge estimation for often used performance metrics in Earth System modeling, refer to Appendix A, is significant. Hence scientific and decision-making choices based on those metrics for reported discharge should be evaluated with care.

Finally, we encourage knowledge mobilization and further collaboration between the Water Survey of Canada, WSC, the private sector, and universities and research institutes, similar to this work, which will open opportunities for the evaluation of organizational processes and constant improvement and stimulate the need for science improvement.

**Code and data availability**

Data is in the possession of the Water Survey of Canada, WSC, and any access should be arranged by the WSC. Codes can be shared accordingly based on the arrangement and agreement with WSC.

**Author contribution**

SG: Manuscript, coding for data extraction and processing and figure preparation, and conceptualization. PHW: Significant help in writing the manuscript, improvement of figures, and conceptualization. AP: Significant contribution to the manuscript, conceptualization. JF: Initial idea of exploring Canadian hydrometric stations, conceptualization, data and code review, and team management. HL: Contribution to the manuscript



and figures and code review. MPC: Contribution to the manuscript and team manage-
ment.

## Competing interests

At least one of the (co-)authors is a member of the editorial board of Hydrology
and Earth System Sciences.

## Appendix A  Description of Performance Metrics

The performance metrics used in this study to evaluate the difference between re-
constructed discharge values using the proposed standalone Python workflow in this study
and reported discharge values in the WSC operational database are:

1. Runoff ratio, $E_{RR}$, is calculated based on the amount of precipitation that falls
over the period of interest.

$$E_{RR} = \frac{V_Q}{V_P} \tag{A1}$$

in which $V_Q$ and $V_P$ are the volume of the discharge for the station of interest and
precipitation for the upstream area of the station of interest in cubic meters $[m^3]$.
The precipitation volume is based on the ERA5 dataset (Hersbach et al., 2020)
and the upstream area is based on the basin shapefile provided by WSC for ac-
tive hydrometric stations. The remapping of the precipitation to the basin is done
using the EARYMORE python package (Gharari & Knoben, 2021).

2. Nash-Sutcliffe Efficiency, $E_{NSE}$ is calculated based on:

$$E_{NSE} = 1 - \frac{\sum_{t=1}^{N}(Q_{d,t} - Q_{w,i})}{\sum_{t=1}^{N} Q_{d,t} - \bar{Q}_d} \tag{A2}$$

3. Root mean square error, $E_{RMSE}$, is calculated based on:

$$E_{RMSE} = \sqrt{\frac{\sum_{t=1}^{N}(Q_{d,t} - Q_{w,t})^2}{N}} \tag{A3}$$

in which the subscript $d$ represents the discharge from the WSC operational database
and the subscript $w$ represents the discharge that is reconstructed based on the
proposed workflow in this study.

4. Kling-Gupta Efficiency, $E_{KGE}$ is calculated based on:

$$E_{KGE} = 1 - \sqrt{O_1 + O_2 + O_3} \tag{A4}$$

in which the components are:

$$O_1 = (1 - \beta)^2 \tag{A5}$$

$$O_2 = (1 - \alpha)^2 \tag{A6}$$

$$O_3 = (1 - r)^2 \tag{A7}$$

where $\beta$ is the ratio of the mean values ($\beta = \mu_w/\mu_d$), $\alpha$ is the ratio of standard
deviation values ($\alpha = \sigma_w/\sigma_d$), and $r$ is the cross-correlation coefficient value of
discharge from WSC operational database to reconstructed discharge from the work-
flow respectively.





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
