# Peer review of "Exploring the provenance of information across Canadian hydrometric stations: Implications for discharge estimation and uncertainty quantification"

_Hydrology and Earth System Sciences, 2023_

## Author Comment (AC2)

**Cover letter,**

Dear Editor,

Thank you immensely for handling our manuscript submission at HESSD. We greatly value the comments received by experts in the field.

Following is our initial response to the reviewers. A more comprehensive reply, coupled with the manuscript revision, will follow the open discussion phase after your decision.

Both reviewers highlighted the length of the manuscript. While we do our best to address this concern, considering the necessity to elucidate the Canadian hydrometric network, terminologies, and fundamental scientific concepts, a significant reduction in manuscript length might be challenging. We intend to either remove or relocate Figures or Tables that aren't directly aligned with the manuscript's overarching narrative and their related text. However, we'll retain fundamental elements to ensure the message remains accessible to a broader audience that might not have a grounding in discharge estimation methods' basics.

Once again, we thank the editor and reviewers for their engagement in the discussion phase within HESSD.

With kind regards,

Shervan Gharari, on behalf of co-authors, Paul Whitfield, Al Pietroniro, Jim Freer, Hongli Liu, Martyn P. Clark

**Answer to Gemma Coxon, reviewer #1:**

We express our gratitude to the reviewer for providing constructive feedback on our manuscript. Your insightful comments have been instrumental in enhancing the quality of our work.

In our overall response to the reviewers' suggestions regarding relocating parts of the primary explanation to the appendix, we will attempt to do so for figures that might not significantly enhance or detract from the overall understanding (for example, Figures 9, 16, 18, and 19). We might also merge the information in Figures 16 and 17 into one Figure. However, it's crucial to note that not all readers are familiar with rating curves, and their concepts (such as 'shift', a topic mentioned in only a handful of studies, 'temporary shift', etc). We believe these concepts are very interconnected and removing these explanations might hamper non-expert readers' comprehension of the presented study.

This paper details the Water Survey of Canada's standard operating procedures in estimating discharge values from stage values. The paper addresses an important issue that is often not documented and has critical impacts on uncertainties in discharge time series. Generally the paper is well written and the figures are well presented, with lots of interesting examples of different types of rating curves. However, the paper is long with a lot of figures and as a result, the key message of the paper gets lost. I recommend shortening the paper (moving more material to supplementary information) and better clarifying the key aims and messages of the paper in the introduction, conclusions and abstract.

Considering the manuscript's length, our foremost goal was to ensure that hydrologists, especially those less acquainted with streamflow data production methods, could comprehensively understand the complexities involved. Oftentimes, papers only mention the use of a rating curve to calculate discharges from observed stages. However, this oversimplified view may not sufficiently contextualize the practices we aim to convey in our manuscript, particularly when establishing terminologies and regional contexts. We've noted this discrepancy while presenting our work to diverse audiences from various backgrounds. For instance, modelers, who are not from engineering or natural science backgrounds, may possess limited knowledge of discharge estimation processes while fitting and comparing different machine learning models.

To tackle the length issue, we are considering a restructuring of the paper. Potentially, relocating a series of examples to an appendix or a new section would better enable readers to navigate the core sections while opting to explore these examples as needed. This reorganization aims to enhance the manuscript's readability and offer a more focused reading experience.

For example, we would eliminate or move to the appendix Table-1 and 2, Figure-9, 16, 18, and 19.

Abstract – I don't think the abstract is a clear summary of the work that has been conducted and the key messages of the paper. I would recommend revising it to better synthesise the outcomes from the paper.

We thank the reviewer for their comment. We try to emphasize the scientific relevance of this work on residuals and reproducibility first in general and then report the specific findings of this study regarding the station of WSC. We have tried to convey these major points:

1- Terminologies and concepts for a broader audience.

2- Processes of "override" and "temporary shift" in the context of general information.

3- Python workflow to explore and label the period with "override" and "temporary shift".

4- Impact of override and temporary shift; impact on residuals, etc. And the need for new approaches to identity.

**Old abstract:** Accurate discharge values play a critical role in water resource planning and management. However, it is common for users, modelers, and decision-makers to consider these values as true and deterministic, despite the subjective and uncertain nature of the estimation process. To address the issue, this study was conducted to identify the discharge estimation methods and associated uncertainties of hydrometric measurements in Canada. The study involved an exploration of multiple operating procedures for rating curve construction and discharge estimation across 1800 active Water Survey of Canada (WSC) hydrometric stations using an independent workflow. The first step involved understanding the discharge estimation process used by the WSC and the standard operating procedures (SOP) for inferring discharge from stage measurements. During the implementation of the workflow, it was observed that manual intervention and interpretation by hydrographers were required for time-series sequences labeled as "override" and/or "temporary shift". The workflow demonstrated that 67 % of existing records could be adequately recreated following the rating curve and temporary shift concept, while 33 % followed the other discharge estimation methods (override). Novel methods for discharge uncertainty estimation should be sought given the practices of override and temporary shift by the WSC. This study attempts to reconcile the significant issue of estimating uncertainty in published discharge values, particularly in the context of open science and Earth System modeling. By collaborating with the WSC, this research aims to improve the understanding of the processes used for discharge estimation and promote wider access to metadata and measurements for more accurate uncertainty quantification.

**New abstract:** Accurate determination of discharge values forms the bedrock for effective water resource planning and management. Unfortunately, these data are frequently perceived as absolute and deterministic by users, modelers, and decision-makers, despite the inherent subjectivity and uncertainty in the data preparation processes. This study is undertaken to examine the discharge estimation methods utilized by the Water Survey of Canada (WSC) and their impacts on reported discharge values. Firstly, we elucidate the hydrometric station network, essential terminologies, and fundamental concepts of rating curves. Subsequently, we delve into WSC's standard operating procedures (SOPs), including shift, temporary shift, and override in discharge estimation. Based on WSC's records of 1800 active hydrometric stations, we evaluate sample rating curves and their correlation to stage and discharge measurement. We investigate under-ice measurement, ice condition periods frequency, and extreme values in contrast to rating curves. Moreover, employing an independent workflow, we demonstrate that 67% of existing records align with the rating curve and temporary shift concept, while the remaining 33% follow alternative discharge estimation methods (override). Examples from a handful of stations are provided for discharge estimation methods over time. Additionally, we illustrate the impact of override and temporary shifts on commonly assumed uncertainty models. Given the practices of override and temporary shifts within WSC, there is a need to explore innovative methods for discharge uncertainty estimation. We hope

our research helps in the critical challenge of estimating and communicating uncertainty in published discharge values.

L51-52. 'River discharge or streamflow has significant importance for planning, impact and sustainability assessment' – this is very generic and could apply to planning, impact and sustainability assessment of anything! This needs to be more specific to water resources.

Thank you for this comment. We will change this sentence to:

"River discharge or streamflow is the fundamental data upon which hydrology and water management depends."

Aims L99-104 – I find the aims of "the study" quite confusing as it is not clear whether "the study" relates directly to this paper or to a wider project? Please revise this section and more clearly state what your core aims and objectives of this paper are.

We thank the reviewer for this comment. We will rewrite this part as:

"This study seeks to identify critical decisions on discharge estimation processes at the WSC. The study tries to address the following questions:

question 1 to 3.

The response and investigation of the aforementioned questions serve as the foundation for the overarching objectives of standardizing uncertainty quantification and communication within the quality assurance and management system of WSC."

L152-154. What is "discharge activity"? The estimated discharge may then be used to correct what? These sentences are not clear.

The reason for using the "discharge activity" was the JSON key with the same name for discharge measurement in the WSC operational database. We have changed the discharge activity to stage and discharge measurement or simply measurement across the manuscript.

Table 1 and 2. I think you can place these in supplementary information. Many of these terms are described in the text already.

We thank the reviewer for this comment. We will consider moving the tables into the appendix.

Figures 3-5. These are very nice but could you combine these into one figure?

Respectfully, we hold a different perspective from the reviewer. Each of these figures serves a distinct purpose. For instance, Figure 3 delineates a static representation, shifting of rating curves that are often permanent. Meanwhile, Figure 5 illustrates temporary shifts in the rating curve, and Figure 4 demonstrates adjustments made to streamflow/discharge time series, addressing "override" or "temporary shift" scenarios. Initially, our attempt involved consolidating them into a single figure; however, this resulted in significant confusion due to the numerous panels and excessively lengthy captions required for explanation.

L316-317. It would be good to add a sentence here on why you are developing an independent Python workflow.

The reason for the workflow, as highlighted in the abstract, is to identify and label the period of the streamflow with various discharge estimating methods. We try to clarify this further in the text for the revised version.

L371-374. This sentence isn't clear and needs re-writing.

Thank you. We have reworked the sentence.

"Under ice observational points have much lower river discharge in comparison to open water flow for the same stage values and therefore are not used in the construction of rating curves, instead are used to adjust the estimated discharge using override values or temporary shifts during the ice condition (Figure 6c). "

"Under a winter ice cover, discharges are lower than during open water and measurements often do not fall on the stage-discharge curve. Instead, while ice is present, the observations are used to adjust the estimated discharges using overrides or temporary shifts (Figure 6c).

L405. The Environment Agency for England does not use this method. They use this method: https://assets.publishing.service.gov.uk/government/uploads/system/uploads/attachment_data/file/29 0629/sw6-058-tr-e-e.pdf

Thank you very much. We will correct the reference about the methodology.

L411-418. The text on observed stage-discharge records is out of place here. It could be removed.

We thank the reviewer. We may move this to the earlier section of the explanation on the bullet points for under-ice conditions.

Figure-6 and 7 – you could move some of these examples to the supplementary information and combine these different examples of rating curves?

Referring to our overall response to the reviewer, we strongly believe in the significance of retaining these examples. They essentially illustrate the deterministic process involved in creating rating curves, a process that varies among hydrographers, offices, and even from year to year. Emphasizing the impact of ice in this context is crucial as it leads to a reduction in the available points for constructing the rating curve. Additionally, these figures are needed to link/explain Figures 16 and 17 of the manuscript.

Figure 10. I like this figure a lot and really interesting to see the regional differences.

We thank the reviewer for that. We're required to implement minor adjustments as per the request from the Copernicus office to enhance the figure's accessibility for individuals with color blindness.

L488-494. The description of the figure can be moved into the figure caption.

We appreciate the reviewer's input. While this explanation exists within the figure's caption, given that it initiates a sequence of four consecutive figures, we opt to explicitly introduce the panel to acquaint the reader and enhance the overall flow. This approach aims to facilitate smoother comprehension throughout the subsequent four figures, even though the caption covers similar details.

Also, we would like to mention that we will rearrange Figures 11 to 14. We start with the simplest case which is currently Figure 13 and move to a more complex case which is Figure 14.

L496. "significantly lower" – can you quantify this? How much lower?

We intended to convey substantial differences, several times in magnitude, between the estimated discharge by rating curve only and under ice discharge estimation by the WSC. To enhance clarity, we'll either eliminate "significantly" or rephrase the sentences for better communication.

Discussion and Conclusions – I would recommend splitting these and having a separate conclusions section where you turn your bullet points in L735-761 into a conclusions section.

We thank the reviewer for this suggestion. We do our best to split the section into two parts. Perhaps the section will be renamed as "Results and Discussions", "Implications", and "Conclusions" or "Results", "Discussion", and "Conclusions".

Data availability. I appreciate that the streamflow data would need to be requested from the WSC but are there any other outputs from your extensive analysis that could be made available to users? For example, could you release the fraction of the discharge within 5% of reported discharge values for each station, or the number of days with a temporary shift for each station, or the fraction of time higher than the maximum observed stage? These outputs could be valuable for researchers conducting large-sample studies in Canada and could be used as a (admittedly crude) way of filtering out stations with more/less robust data.

The reviewer has raised a very valid point. As authors of this work, we've explored the prospect of disseminating the data and its analysis into the public domain. However, a significant hurdle lies in the legal implications associated with assessing data quality. WSC Canada strictly adheres to Standard Operating Procedures (SOPs) for estimating discharge values, ensuring these values are legally defensible. In this study, the Python workflow lacks the comprehensive details of SOPs that WSC follows (such as how temporary shift magnitude is estimated or how overrides are applied).

For future studies, we aim to assess a handful of stations among the entire network, comparing them with in-office information. This approach will provide a more comprehensive understanding of station practices and details specific to individual stations. Notably, the current study primarily focuses on overarching practices with illustrative examples that avoid excessive specificity.

Our Python workflow could be shared with other scientists, contingent upon obtaining the necessary permissions from WSC to utilize the data.

Once again, we would like to thank the reviewer for their constructive comments.

With kind regards,

Shervan Gharari, on behalf of co-authors, Paul Whitfield, Al Pietroniro, Jim Freer, Hongli Liu, Martyn P. Clark

---

## Author Response (AR1)

**Cover letter,**

Dear Editor,

Thank you immensely for handling our manuscript submission at HESSD. We greatly value the insightful comments from the reviewers.

Below is our comprehensive response, along with the revised manuscript. Both reviewers noted the length of the manuscript. While we tried to address this concern, a significant reduction is challenging due to the necessity of explaining the Canadian hydrometric network, terminologies, and foundational scientific concepts. We have removed or relocated figures and tables that were not directly aligned with the manuscript's main narrative and adjusted the related text accordingly. However, we have retained essential elements to ensure the content remains accessible to a broader audience that may not be familiar with the basics of discharge estimation methods. For example, it is very challenging to explain the concept of shift or temporary shift without explaining the concept of rating curve.

We apologize for the delay in preparing the revision. We encountered an unexpected failure in the Canadian High-Performance Computer storage hardware which affected all the files that we have prepared for his manuscript.

Once again, we thank the editor and reviewers for their engagement during the discussion phase within HESSD.

Warm regards,

Shervan Gharari, on behalf of co-authors; Paul Whitfield, Alain Pietroniro, Jim Freer, Hongli Liu, and Martyn P. Clark

**Answer to Gemma Coxon, reviewer #1:**

We express our gratitude to the reviewer for providing constructive feedback on our manuscript. Your insightful comments enhanced the quality of our work.

In our overall response to the reviewers' suggestions regarding relocating parts of the primary explanation to the appendix, we merged and moved Table 1 and Table 2 in the original manuscript to the appendix. We also removed Figures 18 and 19 and its related text. We also replaced Figures 16 and 17 with new Figures that better explain the impact of discharge estimation processes on residuals. However, it's crucial to note that not all readers are familiar with rating curves, and their concepts (such as 'shift', a topic mentioned in only a handful of studies, 'temporary shift', etc). These concepts are very interconnected and removing these explanations might hamper non-expert readers' comprehension of the presented study.

This paper details the Water Survey of Canada's standard operating procedures in estimating discharge values from stage values. The paper addresses an important issue that is often not documented and has critical impacts on uncertainties in discharge time series. Generally the paper is well written and the figures are well presented, with lots of interesting examples of different types of rating curves. However, the paper is long with a lot of figures and as a result, the key message of the paper gets lost. I recommend shortening the paper (moving more material to supplementary information) and better clarifying the key aims and messages of the paper in the introduction, conclusions and abstract.

Considering the manuscript's length, our foremost goal was to ensure that hydrologists, especially those less acquainted with streamflow data production methods, could comprehensively understand the complexities involved. Oftentimes, papers only mention the use of a rating curve to calculate discharges from observed stages. However, this oversimplified view may not sufficiently contextualize the practices we aim to convey in our manuscript, particularly when establishing terminologies and regional contexts. We've noted this discrepancy while presenting our work to diverse audiences from various backgrounds. For instance, modelers, who are not from engineering or natural science backgrounds, may possess limited knowledge of discharge estimation processes while fitting and comparing different machine learning models.

To tackle the length issue, we moved and merged Tables 1 and 2 to the appendix and removed the original Figures 18 and 19. We believe the rest of the manuscript is essential for a non-expert reader to fully comprehend the discharge estimation processes. This reorganization aims to enhance the manuscript's readability and offer a more focused reading experience.

Abstract – I don't think the abstract is a clear summary of the work that has been conducted and the key messages of the paper. I would recommend revising it to better synthesise the outcomes from the paper.

We thank the reviewer for their comment. We tried to emphasize the scientific relevance of this work on residuals and reproducibility first in general and then report the specific findings of this study regarding the station of WSC. We have tried to convey these major points:

1- Terminologies and concepts for a broader audience.

2- Processes of "override" and "temporary shift" in the context of general information.

3- Python workflow to explore and label the period with "override" and "temporary shift".

4- Impact of override and temporary shift; impact on residuals, etc. And the need for new approaches to identity.

**Old abstract:** Accurate discharge values play a critical role in water resource planning and management. However, it is common for users, modelers, and decision-makers to consider these values as true and deterministic, despite the subjective and uncertain nature of the estimation process. To address the issue, this study was conducted to identify the discharge estimation methods and associated uncertainties of hydrometric measurements in Canada. The study involved an exploration of multiple operating procedures for rating curve construction and discharge estimation across 1800 active Water Survey of Canada (WSC) hydrometric stations using an independent workflow. The first step involved understanding the discharge estimation process used by the WSC and the standard operating procedures (SOP) for inferring discharge from stage measurements. During the implementation of the workflow, it was observed that manual intervention and interpretation by hydrographers were required for time-series sequences labeled as "override" and/or "temporary shift". The workflow demonstrated that 67 % of existing records could be adequately recreated following the rating curve and temporary shift concept, while 33 % followed the other discharge estimation methods (override). Novel methods for discharge uncertainty estimation should be sought given the practices of override and temporary shift by the WSC. This study attempts to reconcile the significant issue of estimating uncertainty in published discharge values, particularly in the context of open science and Earth System modeling. By collaborating with the WSC, this research aims to improve the understanding of the processes used for discharge estimation and promote wider access to metadata and measurements for more accurate uncertainty quantification.

**New abstract:** Accurate determination of discharge values forms the bedrock for effective water resource planning and management. Unfortunately, these data are frequently perceived as absolute and deterministic by users, modelers, and decision-makers, despite the inherent subjectivity and uncertainty in the data preparation processes. This study is undertaken to examine the discharge estimation methods utilized by the Water Survey of Canada (WSC) and their impacts on reported discharge values. Firstly, we elucidate the hydrometric station network, essential terminologies, and fundamental concepts of rating curves. Subsequently, we delve into WSC's standard operating procedures (SOPs), including shift, temporary shift, and override in discharge estimation. Based on WSC's records of 1800 active hydrometric stations, we evaluate sample rating curves and their correlation to stage and discharge measurement. We investigate under-ice measurement, ice condition periods frequency, and extreme values in contrast to rating curves. Moreover, employing an independent workflow, we demonstrate that 69% of existing records align with the rating curve and temporary shift concept, while the remaining 31% follow alternative discharge estimation methods (override). Examples from a handful of stations are provided for discharge estimation methods over time. Additionally, we illustrate the impact of override and temporary shifts on commonly assumed uncertainty models. Given the practices of override and temporary shifts within WSC, there is a need to explore innovative methods for discharge uncertainty estimation. We hope our research helps in the critical challenge of estimating and communicating uncertainty in published discharge values.

L51-52. 'River discharge or streamflow has significant importance for planning, impact and sustainability assessment' – this is very generic and could apply to planning, impact and sustainability assessment of anything! This needs to be more specific to water resources.

Thank you for this comment.  We will change this sentence to:

"River discharge or streamflow is the fundamental data upon which hydrology and water management depends."

Aims L99-104 – I find the aims of "the study" quite confusing as it is not clear whether "the study" relates directly to this paper or to a wider project? Please revise this section and more clearly state what your core aims and objectives of this paper are.

We thank the reviewer for this comment. We will rewrite this part as:

"This study seeks to identify critical decisions on discharge estimation processes at the WSC. The study tries to address the following questions:

question 1 to 3.

The response and investigation of the aforementioned questions serve as the foundation for the overarching objectives of standardizing uncertainty quantification and communication within the quality assurance and management system of WSC."

""

L152-154. What is "discharge activity"? The estimated discharge may then be used to correct what? These sentences are not clear.

The reason for using the "discharge activity" was the JSON key with the same name for discharge measurement in the WSC operational database. We have changed the discharge activity to stage and discharge measurement or simply measurement across the manuscript.

Table 1 and 2. I think you can place these in supplementary information. Many of these terms are described in the text already.

We thank the reviewer for this comment. We merged and moved the tables into the appendix.

Figures 3-5. These are very nice but could you combine these into one figure?

Respectfully, we hold a different perspective from the reviewer. Each of these figures serves a distinct purpose. For instance, Figure 3 delineates a static representation, shifting of rating curves that are often permanent. Meanwhile, Figure 5 illustrates temporary shifts in the rating curve, and Figure 4 demonstrates adjustments made to streamflow/discharge time series, addressing "override" or "temporary shift" scenarios. Initially, our attempt involved consolidating them into a single figure; however, this resulted in significant confusion due to the numerous panels and excessively lengthy captions required for explanation.

L316-317. It would be good to add a sentence here on why you are developing an independent Python workflow.

The reason for the workflow, as highlighted in the abstract, is to identify and label the period of the streamflow with various discharge estimating methods. We tried to clarify this further in the text (the second and third sentences of the paragraph of section 2.6). Is that what the reviewer is asking?

L371-374. This sentence isn't clear and needs re-writing.

Thank you. We have reworked the sentence.

"Under ice observational points have much lower river discharge in comparison to open water flow for the same stage values and therefore are not used in the construction of rating curves, instead are used to adjust the estimated discharge using override values or temporary shifts during the ice condition (Figure 6c). "

"Under a winter ice cover, discharges are much lower than during open water and measurements often do not fall on the stage-discharge curve. Instead, while ice is present, the observations are used to adjust the estimated discharges using overrides or temporary shifts (Figure 6c).

L405. The Environment Agency for England does not use this method. They use this method: https://assets.publishing.service.gov.uk/government/uploads/system/uploads/attachment_data/file/290629/sw6-058-tr-e-e.pdf

Thank you very much. We corrected the reference about the methodology.

L411-418. The text on observed stage-discharge records is out of place here. It could be removed.

We thank the reviewer. We remove this extra text.

Figure-6 and 7 – you could move some of these examples to the supplementary information and combine these different examples of rating curves?

Referring to our overall response to the reviewer, we strongly believe in the significance of retaining these examples. They essentially illustrate the deterministic process involved in creating rating curves, a process that varies among hydrographers, offices, and even from year to year. Emphasizing the impact of ice in this context is crucial as it leads to a reduction in the available points for constructing the rating curve. Additionally, these figures are needed to link/explain Figures 16 and 17 of the revised manuscript.

Figure 10. I like this figure a lot and really interesting to see the regional differences.

We thank the reviewer for that. We're required to implement minor adjustments as per the request from the Copernicus office to enhance the figure's accessibility for individuals with color blindness.

L488-494. The description of the figure can be moved into the figure caption.

We appreciate the reviewer's input. While this explanation exists within the figure's caption, given that it initiates a sequence of four consecutive figures, we opt to explicitly introduce the panel to acquaint the reader and enhance the overall flow. This approach aims to facilitate smoother comprehension throughout the subsequent four figures, even though the caption covers similar details.

L496. "significantly lower" – can you quantify this? How much lower?

We intended to convey substantial differences, several times in magnitude, between the estimated discharge by rating curve only and under ice discharge estimation by the WSC. To enhance clarity, we eliminated "significantly" for better communication.

Discussion and Conclusions – I would recommend splitting these and having a separate conclusions section where you turn your bullet points in L735-761 into a conclusions section.

We thank the reviewer for this suggestion. We split the section into two parts: "Discussions", and "Conclusions".

Data availability. I appreciate that the streamflow data would need to be requested from the WSC but are there any other outputs from your extensive analysis that could be made available to users? For example, could you release the fraction of the discharge within 5% of reported discharge values for each station, or the number of days with a temporary shift for each station, or the fraction of time higher than the maximum observed stage? These outputs could be valuable for researchers conducting large-sample studies in Canada and could be used as a (admittedly crude) way of filtering out stations with more/less robust data.

The reviewer has raised a very valid point. As authors of this work, we've explored the prospect of disseminating the data and its analysis into the public domain. However, a significant hurdle lies in the legal implications associated with assessing data quality. WSC Canada strictly adheres to Standard Operating Procedures (SOPs) for estimating discharge values, ensuring these values are legally defensible. In this study, the Python workflow lacks the comprehensive details of SOPs that WSC follows (such as how temporary shift magnitude is estimated or how overrides are applied).

For future studies, we aim to assess a handful of stations among the entire network, comparing them with in-office information. This approach will provide a more comprehensive understanding of station practices and details specific to individual stations. Notably, the current study primarily focuses on overarching practices with illustrative examples that avoid excessive specificity.

Our Python workflow could be shared with other scientists, contingent upon obtaining the necessary permissions from WSC to utilize the data.

Finally, we would like to thank the reviewer for their constructive comments on our work that enriched our manuscript.

With kind regards,

Shervan Gharari, on behalf of co-authors

**Answer to Anonymous Reviewer, Reviewer #2:**

This article presents a systematic study of the hydrometric data production process across 1800 active stations operated by Water Survey Canada. An independent (Python-based) approach intended to reconstruct the archived discharge times series based on information and data available from the Aquarius operational software. Interestingly, only 67% of the data could be reproduced (within 5%) from the stage series, the rating curves, and the rating shift curves, the other differences being explained by the significant use of temporary shifts and "overrides". This exercise is valuable as it quantifies the frequency of operational practices that are more complex to reproduce than the simple application of rating curves (and permanent shifts). In particular, the need for suitable uncertainty computation methods is rightly emphasized.

The paper is generally very well written and well illustrated, however, I fear that its length may discourage some readers less passionate about hydrometry (including data users!) and reduce its impact. I would recommend shortening the paper (20 pages max and 10 figures max). Some technical details (eg multiple data examples) could be cut or moved to Annexes or Supplementary materials.

We thank the reviewer for their positive and constructive feedback on this work.

Similar to our response to editor and reviewer #1's feedback, we aim to relocate specified material from the main text to the appendix, consolidating figures to reduce their count. While we will push to limit the figures, we cannot guarantee a maximum cap.

Both reviewers demonstrate expertise in the field, suggesting that detailed explanations of rating curves or fundamental concepts may be redundant. However, considering a broader audience, establishing foundational knowledge remains crucial. It's been observed that colleagues utilizing discharge for data-driven modeling, such as machine learning, may lack familiarity with rating curves. Recognizing this gap, the first author noted, when doing the literature review, that many works on hydrometric stations tend to lose connection with a wider audience, due to deep diving directly into detailed technicalities.

69-70 the method (IVE) introduced by Cohn et al. 2013 does not relate to rating curves. Not sure about Whalley et al. 2001 and Huang 2018. Please check and remove if need be.

We thank the reviewer, and we reformulated the paragraphs and corrected the references. It seems due to the changes in the text, publications related to measurement uncertainty and rating curve uncertainty were mixed.

405 I'm not sure the method presented by Coxon et al. 2015 is actually applied systematically by UK Env agency to establish their rating curves. I don't think so. Kiang et al. 2018 compared 7 methods for rating curve uncertainty and only the NVE method (in Norway) and the Baratin method (in France) were applied by national hydrological services.

We thank the reviewer, and we corrected the reference to UK SOP for rating curve creation (also per reviewer #1 request)

What about (seasonal) aquatic vegetation? Is it a problem for Canadian stations (as it is in Europe for instance) and is it managed through temporary shifts? I assume that beaver dams are another issue…

The correlation between shifts and seasons frequently follows a distinct pattern. Colder periods often experience more pronounced intervention from processes like "overrides" and "temporary shifts." Interestingly, in our limited exploration, we found a lack of discussion on vegetation within the operational database, assuming these data have been accurately transferred to the digital operational database.

Additionally, the impact of beavers tends to be localized over smaller areas. While we observed the beaver effect on some experimental catchments, the stations managed by WSC typically cover larger river segments and tributaries.

As stated end of 2.5 and elsewhere (Tab. 3), the central issue is the traceability, reproducibility of the data production process. However, reproducibility and repeatability are different things, and this could be made clearer in the paper. A first step is that discharge computation can be repeated (exactly) using available data and already established rating curves, shifts and overrides (from Aquarius especially): this doesn't seem to be the case as some important information is missing (or not easily retrieved through API), which would be a first issue of incomplete traceability (am I correct?). Another step is that discharge computation can be reproduced (with some permissible variability) from scratch by another equivalent expert: this should be OK thanks to established SOPs and well-trained operators, hopefully, but this statement is not evaluated in this study (through some comparison exercise, for instance). Actually, the problem seems to arise because the assumptions and decisions made by the hydrographers for establishing rating curves, rating shifts, temporary shifts and overrides are not available in a formal way. I think that beyond the statistical technique chosen for uncertainty estimation, this is the key issue: each data production process must be 'modelled' in a reproducible way, even expert-based operations. I agree that some solutions have been published that apply to rating curves and (partially) shifts but not to temporary shifts and overrides. This paper is a first step towards modelling these operations but much more work looks necessary to write mathematical models, especially for 'override' operations, which refer to multiple situations and data estimation techniques. The discussion and comments in the paper could elaborate on this issue more clearly.

This response effectively captures our intended message. However and initially, we refrain from delving deeply into the detailed explanations of reproducibility and repeatability. While Standard Operating Procedures (SOPs) and training ensure the repeatability of results, achieving reproducibility demands a deeper understanding and an underlying explanatory model. This model can serve in estimating discharge by non-experts and form the basis for uncertainty analysis, akin to uncertainty models for rating curves.

To emphasize these points, we incorporated a discussion paragraph addressing the reviewer's comment on the distinction between reproducibility and repeatability.

Another obstacle stated in the paper is the deterministic approach: the uncertainty of stage-discharge measurements must be accounted for, as well as the uncertainty of the input data (stage) and of the rating curves (and more generally the "discharge models"). It looks difficult to quantify the uncertainty of data that have been produced in a fully deterministic approach without reprocessing

them. The ideal way to go is to reproduce the data in a probabilistic framework, hence the need for reproducibility…

The reviewer's perspective aligns with our initial project goal, which centered around streamflow uncertainty assessment. Originally, our focus was on contemplating potential uncertainties in streamflow, encompassing stage, measurement, and rating curve uncertainties. However, as the authors delved deeper into WSC practices, the project's focus shifted towards comprehending existing processes rather than solely estimating uncertainty. The process of uncertainty estimation necessitates an explanatory model, which, to our current knowledge, remains absent. Without this foundational model, quantifying uncertainty would be exceptionally challenging. We hope this work is a step toward that "model".

146 Aquatic? Corrected from Aquatics

176 include? Corrected from includes

230 curve corrected from curved

558 To investigate what? This section is fully removed in the revision

738 Pytho corrected to "Python"

We would like to thank the reviewer for their constructive comments on our work that enriched our manuscript.

With kind regards,

Shervan Gharari, on behalf of co-authors